# Perturbation of METTL1-mediated tRNA N[7]-methylguanosine modification induces senescence and aging

Yudong Fu[1,2,3,11,12], Fan Jiang[1,2,3,12], Xiao Zhang[1,2,3,11], Yingyi Pan[4], Rui Xu[5], Xiu Liang[1,2,3], Xiaofen Wu[1,2,3], Xingqiang Li[6], Kaixuan Lin[1,2,3], Ruona Shi[1,2,3], Xiaofei Zhang [1,2,3,11], Dominique Ferrandon[4,7,8], Jing Liu [1,2,3,9,11], Duanqing Pei [10], Jie Wang [1,2,3,9,11] ✉ & Tao Wang [1,2,3,11] ✉

Cellular senescence is characterized by a decrease in protein synthesis, although the underlying processes are mostly unclear. Chemical modifications to transfer RNAs (tRNAs) frequently influence tRNA activity, which is crucial for translation. We describe how tRNA N7-methylguanosine (m7G46) methylation, catalyzed by METTL1-WDR4, regulates translation and influences senescence phenotypes. Mettl1/Wdr4 and m7G gradually diminish with senescence and aging. A decrease in METTL1 causes a reduction in tRNAs, especially those with the m7G modification, via the rapid tRNA degradation (RTD) pathway. The decreases cause ribosomes to stall at certain codons, impeding the translation of mRNA that is essential in pathways such as Wnt signaling and ribosome biogenesis. Furthermore, chronic ribosome stalling stimulates the ribotoxic and integrative stress responses, which induce senescence-associated secretory phenotype. Moreover, restoring eEF1A protein mitigates senescence phenotypes caused by METTL1 deficiency by reducing RTD. Our findings demonstrate that tRNA m7G modification is essential for preventing premature senescence and aging by enabling efficient mRNA translation.

Cell senescence is characterized by permanent growth arrest while remaining metabolically active[1]. Accumulation of senescent cells in tissues causes organism aging[2–5]. It has been reported that protein synthesis capacity decreases with senescence and aging[6–8]. The mechanism causing a reduced protein translation in senescence is still not well understood. Protein synthesis is an energy and resource-consuming biological process. Normal cells delicately coordinate the process to adapt to the changing microenvironments. During protein synthesis, transfer RNAs (tRNAs) act as adapters, transporting amino acids to nascent peptide chains, enabling the completion of the protein synthesis. tRNA transcripts are typically small non-coding RNAs, around 70-90 nucleotides long. tRNAs are transcribed by RNA polymerase III as longer precursor molecules and processed into mature tRNAs through several steps, including splicing, 3' end CCA addition,

[1]Guangdong Provincial Key Laboratory of Stem Cell and Regenerative Medicine, Guangdong-Hong Kong Joint Laboratory for Stem Cell and Regenerative Medicine, Guangzhou Institutes of Biomedicine and Health, Chinese Academy of Sciences, Guangzhou, China. [2]GIBH-HKU Guangdong-Hong Kong Stem Cell and Regenerative Medicine Research Centre, Guangzhou, China. [3]GIBH-CUHK Joint Research Laboratory on Stem Cell and Regenerative Medicine, Guangzhou, China. [4]Sino-French Hoffmann Institute, Guangzhou Medical University, Guangzhou, China. [5]Department of pediatrics, Foshan maternal and children's hospital affiliated to southern medical university, 528000 Foshan, Guangdong, China. [6]Bioland Laboratory, Guangzhou, China. [7]Université de Strasbourg, Strasbourg, France. [8]Modèles Insectes de l'Immunité Innée, UPR 9022 du CNRS, Strasbourg, France. [9]Joint School of Lifesciences, Guangzhou Institutes of Biomedicine and Health, Chinese Academy of Sciences, Guangzhou, 510530, China, Guangzhou Medical University, 511436 Guangzhou, China. [10]School of Life Sciences, Westlake University, Hangzhou, China. [11]University of Chinese Academy of Sciences, Beijing, China. [12]These authors contributed equally: Yudong Fu, Fan Jiang. ✉e-mail: wang_jie01@gibh.ac.cn; wang_tao@gibh.ac.cn

and chemical modifications. Mature tRNAs exhibit a cloverleaf and tertiary L-shaped structure[9]. Amino acids must attach to tRNA for protein synthesis, which is catalyzed by specific aminoacyl tRNA synthetases. In addition to tRNA aminoacylation, various base modifications play a crucial regulatory role in tRNA activity[10,11]. More than a dozen RNA modifications are believed to be present in tRNAs at various positions, introduced by specific enzymes[12,13]. The reactions are classified into methylation, pseudouridylation, sulfuration, and other modifications[13–19]. Depending on their positions and types, chemical modifications in tRNAs play various regulatory roles, influencing the interaction between cognate codons and anticodons, tertiary structure folding, and thermal stability[11,20–22]. tRNA chemical modifications and their abundance are dynamically regulated in response to the cellular metabolic state and environmental stress[23–25]. Certain tRNA chemical modifications play a role in regulating various biological functions. For instance, METTL1-mediated m7G methylation of tRNA at the G46 site is upregulated in multiple tumors[26–28]. The lack of m7G modification causes neurodevelopmental defects[29], and alters embryonic stem cell differentiation[30]. The m7G modification at position 46 of tRNAs is conserved in prokaryotes, eukaryotes, and some archaea[31,32]. This modification is recognized as an important mechanism to prevent tRNAs from degradation, thereby ensuring efficient protein translation[33]. The heterodimer of METTL1-WDR4 is responsible for adding a methylation group to guanine in RNAs. The interrogation of the METTL1-WDR4-tRNA complex structure has enhanced our understanding of m7G methylation mechanisms[34,35]. Dysregulation of tRNA modifications is commonly observed in various pathologies, such as tumors and age-related diseases like Alzheimer's and Parkinson's diseases, which are characterized by abnormal protein translation[11,36–39]. These clues raise questions about the potential role of tRNA modifications in regulating abnormal protein translation during senescence and aging.

Here, we demonstrate that m7G tRNA methylation is crucial for preventing premature senescence and aging by maintaining efficient mRNA translation. METTL1 and WDR4 are downregulated during cell senescence and aging at both the transcriptional and protein levels. This results in a subset of tRNAs being targeted for RTD degradation due to m7G46 hypomethylation. Knockdown of METTL1 drives proliferating cells into senescence and significantly shortens the lifespan of mice. Overexpression of METTL1 delays cell senescence and mitigates tissue damage induced by chemotherapy agents. METTL1 deficiency leads to ribosome stalling at specific codons, triggering ribotoxic stress response(RSR) and integrated stress response(ISR). This suppresses global translation initiation and elongation while activating the senescence-associated secretory phenotype (SASP). Conversely, the restoration of METTL1 alleviates senescence-associated proteostasis collapse by maintaining efficient mRNA translation. This study demonstrates the important role and underlying molecular mechanism of METTL1-WDR4 and m7G tRNA modification in senescence. The finding highlights the potential of this intervention for promoting healthy aging and treating age-related diseases.

## Results

### RNA m7G methylation decreases with senescence and aging
Chemical modifications of RNAs are catalyzed by various enzymes. We aimed to identify differentially expressed RNA modification complexes between proliferating and senescent IMR90 cells (primary human fetal lung fibroblasts) using a proteomics assay. Several protein complexes involved in RNA modifications, such as N6-methyladenosine (m6A), 5-methylcytosine (m5C), and N7-methylguanosine (m7G), were observed to change during replicative senescence (Fig. 1A, Supplementary Data 1). Previous reports indicated that protein translation was a significant biological activity of the METTL1-WDR4 complex[30].

Therefore, we have chosen this complex to investigate how the RNA m7G methylation-mRNA translation axis regulates senescence.

We showed that METTL1-WDR4 was downregulated in three senescence models, replicative senescence (RS), therapy-induced senescence (TIS), and oncogene-induced senescence (OIS), through Western blotting and RT-qPCR (Fig. 1B, C, Fig. S1A). Moreover, the protein levels of METTL1 and WDR4 gradually decreased during replicative senescence in IMR90 cells, correlating with their division capacity (Fig. 1D, Fig. S1B). Mettl1 and Wdr4 were also notably downregulated in aged mouse liver, kidney and lung tissues (Fig. 1E, Fig. S1C, Fig. S1D). The m7G levels in total RNA, tRNA, and mRNA significantly decreased in senescent cells, aged mouse tissues (liver and kidney) as measured by HPLC-MS (Fig. 1F, G). This trend reflected changes in METTL1 and WDR4. Given the decrease in METTL1-WDR4 levels and m7G methylation during senescence and aging, there is a growing hypothesis that METTL1-WDR4 may regulate senescence.

### METTL1 regulates senescence
To explore the potential connection between METTL1-WDR4 and senescence, we performed loss-of-function and gain-of-function experiments in IMR90 cells. We utilized two different short hairpin RNAs (shRNAs) to effectively reduce METTL1 expression. It was found that senescence-related proteins p16[INK4a] and p21[CIP1] were gradually upregulated, while EZH2 and LMNB1 were downregulated in the METTL1 knockdown cells (Fig. 2A). The 5-ethynyl-2'-deoxyuridine (EdU) incorporation assay showed that METTL1-depleted cells ceased the division over time (Fig. 2B). After twelve days of depleting METTL1, there was a significant rise in senescence-associated β-galactosidase (SA-β-gal) positive cells with flat and enlarged nuclei, the feature for senescent cells[40](Fig. 2C). To validate the phenotypes, we used CRISPR-Cas9 guide RNAs (sgRNAs) to knock out METTL1. The results from various assays, such as senescence-associated protein expression levels, SA-β-gal staining, EdU incorporation, and KEGG analysis of transcriptome showed that METTL1 knockout induced cellular senescence (Fig. 2D, S2A, B, C). Moreover, the SASP genes also exhibited the enrichment in the differentially regulated genes upon METTL1 knockout (Fig. S2D, Supplementary Data 2). Consistently, the m7G modification on total RNA and tRNA decreased in METTL1-depleted cells, as quantified by HPLC-MS (Fig. S2E and, Supplementary Data 7).

Furthermore, KO-rescue experiments were conducted through the overexpression of vector control (GFP), METTL1, and AFPA-mutant in METTL1 knockout cells. The examination included the evaluation of senescence markers, cell proliferation, and SA-β-gal staining. The findings validated that the wildtype METTL1, as opposed to the catalytic mutant AFPA-mutant or the control (GFP), effectively reversed senescence phenotypes (Fig. 2E–G). The findings suggest that the absence of METTL1 leads to senescence, relying on m7G catalytic activity.

Based on the data, it is inferred that the preservation of METTL1 levels could enhance RNA m7G levels and delay senescence. To test the assumption, intact METTL1 and a catalytically inactive mutant AFPA-mutant were introduced into pre-senescent IMR90 cells. Stable cell lines were created using antibiotic selection and confirmed for ectopic expression via Western blotting. Two bands were identified using an anti-METTL1 antibody (Fig. S2F). To differentiate the METTL1 protein bands, a FLAG tag was added at the C-terminus (METTL1-FLAG) and a HA tag at the N-terminus (HA-METTL1) (Fig. S2F). Using the METTL1 antibody for both constructs, two bands are visible, and the FLAG antibody also shows two bands (Fig. S2F). But only one band can be detected with the N-terminus HA tag (Fig. S2F, left panels). The upper band may be due to an additional tag sequence added to the native protein. The lower band observed in the results may suggest a truncated protein, possibly due to protease digestion of the N terminus of METTL1.

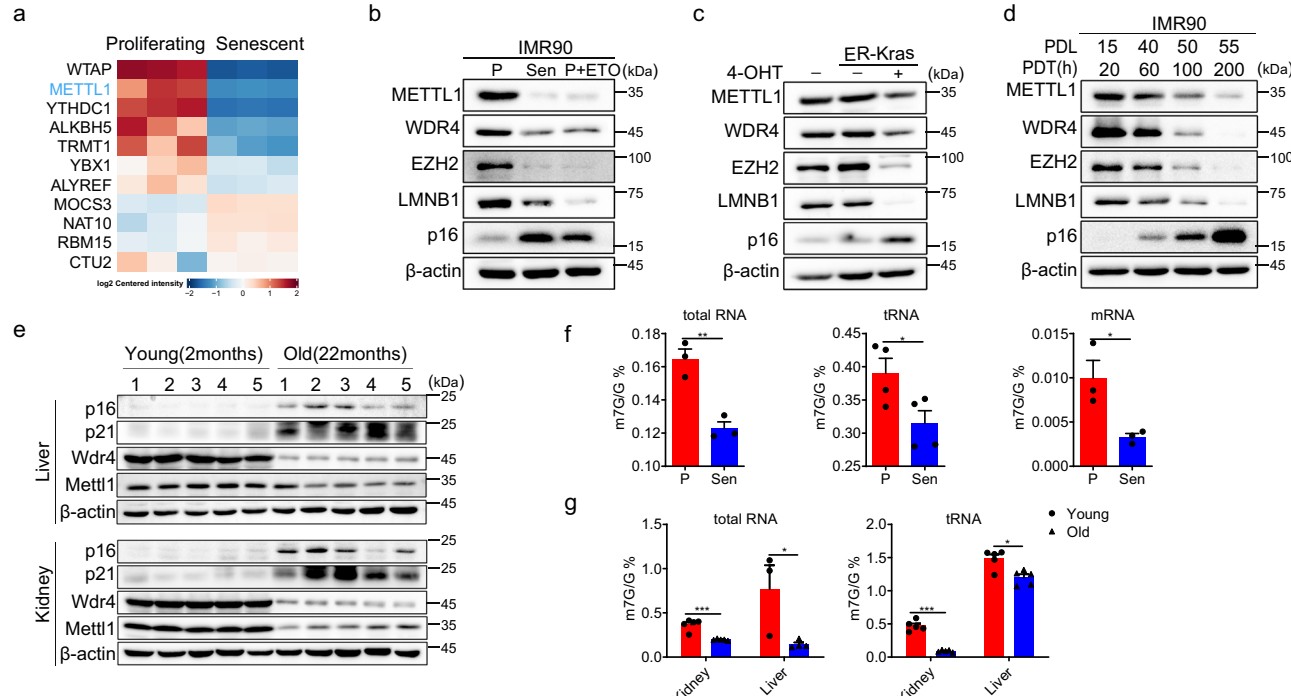

**Fig. 1 | RNA m7G methylation decreases with senescence and aging.**
**a** Differentially expressed RNA-modifying enzymes in IMR90 cells were identified using a proteomics assay. The data was summarized from three repeats. The METTL1/WDR4 m7G modification complex was analyzed by western blotting using RS (replicative senescence) and TIS (therapy-induced senescence) (**b**) and KRasG12V-induced senescence (OIS) (**c**), unless otherwise specified, P represents proliferating IMR90 cells, Sen refers to senescent IMR90 cells. P + ETO stands for proliferating cells treated with ETO. **d** METTL1-WDR4 and senescence markers at different time points during IMR90 cell senescence were detected. **e** Mettl1/Wdr4 were measured by Western blotting in tissues from aged mice ($n = 5$, 2 female, 3 male), specifically in the liver and kidney. m7G levels of total RNA, tRNA, and mRNA were assessed by HPLC-MS in samples of proliferative and senescent IMR90 cells (**f**) and the corresponding mice liver and kidney tissues (**g**). The representative experiment was shown. Three repeats demonstrate similar results (**b, c, d**). Three independent experiments were conducted, unless otherwise specified, $n = 3$, $P$ values were calculated with the two-tailed Student's $t$ test. Data are presented as mean ± SEM, *$p < 0.05$; **$p < 0.01$; ***$p < 0.001$. Source data are provided as a Source Data file.

HPLC-MS confirmed that intact METTL1 overexpression, not the catalytically inactive AFPA-mutant, elevated tRNA m7G levels as measured by HPLC-MS (Fig. S2G). Cells overexpressing METTL1 showed about 35% EdU-positive, while the control (GFP) and AFPA-mutant cells had about 22% (Fig. 2H). We continuously cultured these cell lines in parallel and performed senescence assays when the control cells ceased dividing. SA-β-gal staining showed that METTL1 overexpression significantly decreased the population of senescent cells (Fig. 2I). Intact METTL1 partially restored LMNB1 and EZH2 protein levels and reduced p16$^{INK4a}$ and p21$^{CIP1}$ protein levels in senescent cells (Fig. 2J). The data indicates that restoring METTL1 can delay cell senescence. Similarly, overexpression of METTL1 in etoposide (ETO)-treated IMR90 cells effectively alleviated the damaging effects, reduced the population of SA-β-gal positive cells (Fig. 2K, L), and suppressed p16$^{INK4a}$ and p21$^{CIP1}$ proteins while enhancing LMNB1 and EZH2 protein levels (Fig. 2M). The data indicated that METTL1 overexpression alleviated both replicative senescence and therapy-induced senescence phenotypes. Loss-of-function and gain-of-function studies collectively demonstrate that METTL1 regulates cell senescence in a m7G catalytic activity-dependent manner.

### Depletion of Mettl1 induces premature aging
To determine if METTL1 regulates cell senescence in vivo, we utilized transgenic mice to manipulate Mettl1 expression. The conditional knockout (cKO) and knock-in (cKI) alleles were created by crossing Mettl1$^{flox/flox}$ and Rosa26$^{LSL-Mettl1}$ with the Rosa26$^{CreERT2}$ line, respectively (Fig. S3A, B). 7-8-week-old Rosa26$^{CreERT2}$Mettl1$^{loxp/loxp}$ and Rosa26$^{CreERT2}$ control mice were administered tamoxifen for 7 consecutive days (Fig. 3A). Liver and kidney tissues from conditional knockout (KO) and

control mice were collected three months post-tamoxifen injection to verify recombination efficiency and evaluate m7G levels. It is found that tRNA-m7G levels in METTL1 KO mice were significantly lower compared to those in control mice, as quantified by HPLC-MS (Fig. 3B). The lifespan of Mettl1 knockout mice was significantly reduced to about six months (Fig. 3C). It was found that a significant number of senescent cells accumulated in the liver, kidney tissues of Mettl1 knockout (KO) mice as assessed by SA-β-gal staining (Fig. 3D). The liver, kidney, lung, heart, and small intestine tissues consistently upregulated p16$^{INK4a}$ and p21$^{CIP1}$ compared to control animal tissues (Fig. 3E and S3C–S3F). Mice with lower Mettl1 levels showed signs of degeneration, including thin fur and severe postural kyphosis (Fig. 3F and S3G). In the study, 60% of the knockout (KO) mice developed noticeable lens opacification and progressive cataracts (Fig. 3G, Fig. S3H). Compared to the control group, Mettl1 knockout (KO) mice exhibited significantly poorer muscle performance (Fig. 3H). Cortical bones in Mettl1 knockout (KO) mice also exhibited the reduced volume and thickness compared to control mice (Fig. S3I). The results demonstrated that Mettl1 deficiency significantly accelerated mouse aging. To determine the evolutionary conservation of METTL1's role in aging, we examined the potential effects of CG4045 (the *Drosophila* ortholog of human METTL1) on the lifespan of *Drosophila*. It was observed that the expression levels of CG4045 and Wuho (the *Drosophila* ortholog of human WDR4) declined with age in both male and female *Drosophila* (Fig. S3J). Knockdown of CG4045 using the UAS-GAL4 system led to a notable decrease in lifespan by around 20% (Fig. 3I). The results indicated that METTL1-mediated m7G methylation represented an evolutionarily conserved mechanism in the context of aging.

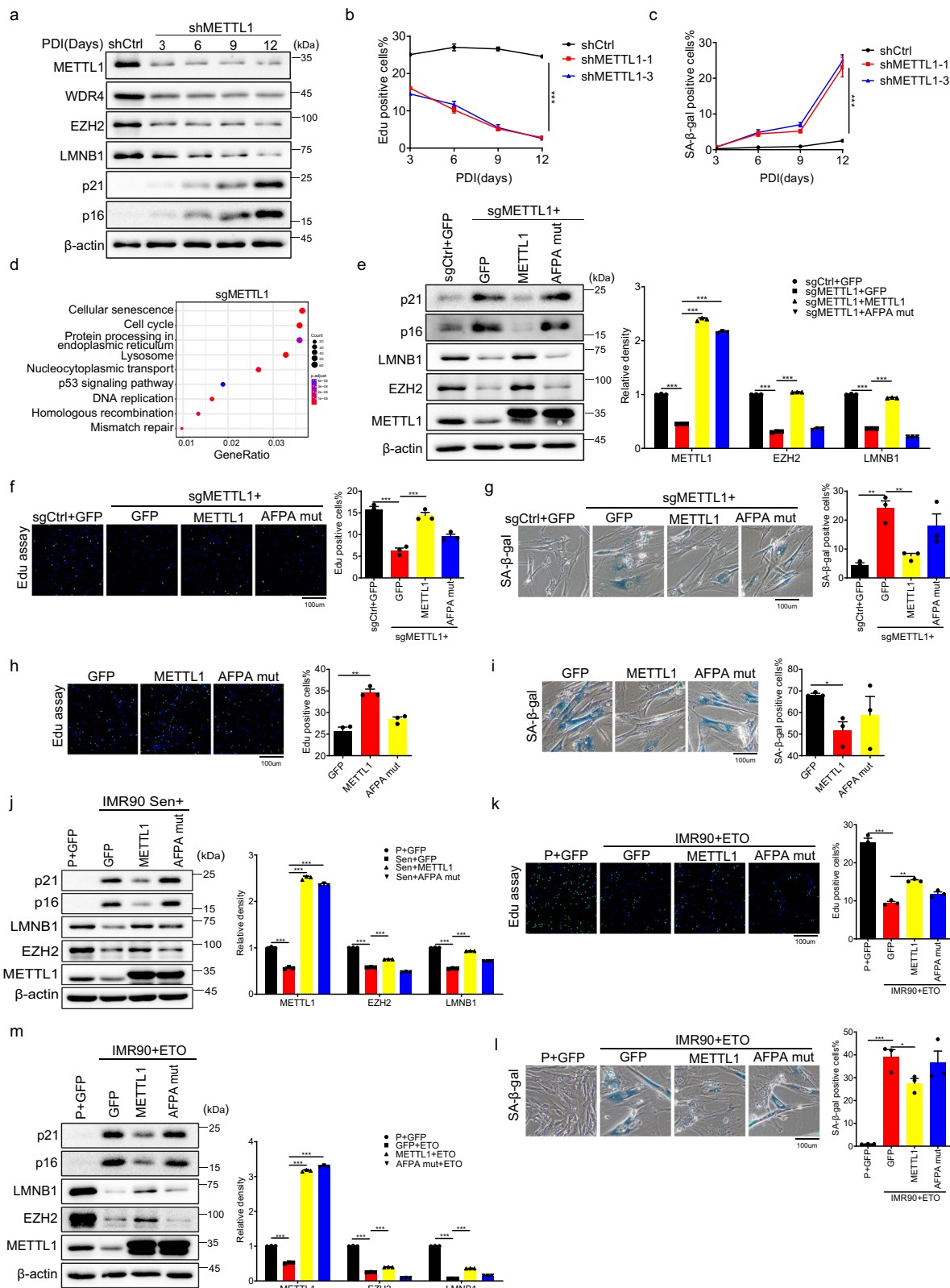

The knockout (KO) result prompted an investigation into whether METTL1 could alleviate senescence in vivo. Hence, a mixture of cis-platin and doxorubicin (CPA) was injected intraperitoneally into Rosa26$^{CreERT2/LSL\text{-}Mettl1}$ mice and control Rosa26$^{CreERT2}$ mice at the doses of 9 mg/kg and 6 mg/kg to induce senescence in the liver and kidney (Fig. 3J). After three injections of CPA, the expression levels of Mettl1 and Wdr4 in the liver were notably reduced (Fig. S3K). We observed that

liver tissues from the group induced by Mettl1 (tamoxifen-treated Rosa26$^{CreERT2/LSL\text{-}Mettl1}$) had a significantly lower percentage of senescent cells compared to the control group (Rosa26$^{CreERT2}$), which showed about four-fold increased intensity of SA-β-gal staining compared to the group of mice with ectopically expressed Mettl1 (Fig. 3K, S3L). Biochemical assays revealed that overexpression of Mettl1 decreased levels of aspartate aminotransferase (AST), alanine aminotransferase

**Fig. 2 | METTL1 regulates senescence. a** Western blotting detected senescence markers at various time points following transduction with METTL1 and control shRNA. A representative experiment was presented, three repeats demonstrate similar results. **b** EdU incorporation assays(**b**) and Senescence-associated β-galactosidase (SA-β-gal) (**c**) staining were conducted on control and METTL1 knockdown cells at various time points after transduction with METTL1 and control shRNA. Scale bar = 100 μm. **d** KEGG analysis was conducted on the transcriptome from METTL1-depleted cells (KO). **e** Senescence-associated proteins were analyzed in METTL1 knockout cells introduced with different proteins. A representative experiment was presented, three repeats demonstrate similar results. The EdU incorporation assay (**f**, **h**) and SA-β-gal staining (**g**, **i**) was performed on METTL1 KO IMR90 cells or wildtype IMR90 cells introduced with indicated proteins. Scale bar = 100 μm. **j** Western blotting for senescence markers was conducted on IMR90 cells transduced with GFP control, METTL1, and AFPA-mutant. P + GFP means proliferating cells with GFP overexpression. Three repeats demonstrate similar results. The representative experiment was shown, band intensity was quantified by image j for three time. ETO treated IMR90 cells expressing METTL1 or AFPA-mutant was assessed by EdU incorporation (**k**), SA-β-gal staining(**l**), and western blotting(**m**), P + GFP refers to IMR90 cells with GFP overexpression. Scale bar = 100 μm. All the assays were biologically repeated three times. All data were presented as the mean ± SEM. $P$ values in panel (**b**, **c**) were calculated by two-tailed unpaired $t$ test, $P$ values in panel (**e**–**m**) were calculated by one-way ANOVA with Bonferroni's multiple comparisons test. *$p < 0.05$; **$p < 0.01$; ***$p < 0.001$. Source data are provided as a Source Data file.

(ALT), serum creatinine (SCr), and blood urea nitrogen (BUN), indicating liver and kidney injury (Fig. 3L). The results suggest that Mettl1 promotes tissue repair and reduces chemotherapy-induced senescence.

The loss-of-function and gain-of-function studies illustrate that METTL1 serves as an important preventive factor against premature senescence and aging, depending on its m7G catalytic activity. Next, we aimed to study the molecular mechanisms of METTL1-mediated senescence. This was crucial for understanding the processes of senescence and aging.

### METTL1 deficiency downregulates a group of m7G-tRNAs

The data shows that METTL1-mediated m7G modification is an essential safeguard against premature senescence both in vitro and in vivo. Due to a substantial amount of m7G methylation originating from tRNAs (Fig. 1F), we hypothesized that tRNAs, being the main substrates for METTL1, were the essential downstream effectors responsible for senescence induced by METTL1 deficiency.

We utilized m7G tRNA reduction and cleavage-sequencing (TRAC-seq), a technique developed to quantify tRNA levels and m7G methylation levels[30], to explore the connection between senescence and alterations in tRNA transcripts and m7G modifications. In the m7G TRAC-Seq procedure, small RNAs were treated with AlkB and D135S mutant proteins to remove blockers inhibiting reverse transcription[41]. Methylated guanosine (m7G) sites were reduced by sodium borohydride (NaBH₄), and followed by specific cleavage at the reduced sites after aniline treatment[42,43]. The tRNA fragments were ligated with 5′ and 3′ terminal adaptors and reverse transcribed using a highly processive thermostable group II intron reverse transcriptase[41,44,45]. cDNA was subjected to high-throughput sequencing using a small RNA library preparation kit to identify m7G methylation sites with high resolution[30,41,45] (Fig. 4A, Fig. S4A and S4B). From the cleavage score data, it was observed that 20 tRNAs in normal IMR90 cells were methylated at their G46 base group (Fig. 4B and Supplementary Data 3). These tRNAs were identified as m7G-tRNAs. The motif identified for m7G sites in these tRNAs was 'GGTTC' following the patten "RRWTM," consistent with a previous report[41] (Fig. 4C). Given that cleavage scores calculated from TRAC-Seq correlated with m7G methylation abundance in a sample, we calculated the ratio of cleavage scores to evaluate m7G methylation changes across samples. We observed that the majority of cleavage score ratios (the control relative to KO) for m7G-tRNAs were above 1 (Fig. 4D), suggesting reduced m7G methylation of these tRNAs in METTL1 KO cells compared to control cells. The results were confirmed by quantifying the abundance of m7G using HPLC-MS (Fig. S2E). We observed that the cleavage score of several tRNAs did not decrease in senescent cells (Fig. S4C), suggesting that other factors beyond METTL1 may regulate the turnover of m7G modification in m7G-tRNAs in senescent cells. The previous study indicated that reducing m7G46 methylation destabilized tRNAs, resulting in decreased abundance[33]. Indeed, the majority of m7G-tRNAs were downregulated in senescent IMR90 cells, as indicated by

an analysis of TRAC-Seq data (Fig. 4E, Supplementary Data 4). The analysis also revealed a gradual downregulation of predominant m7G-tRNAs in cells lacking METTL1 when comparing fold changes of tRNA transcripts on day 6 and day 10 after sgRNA transduction (Fig. 4F, G, and Fig. S4D, Supplementary Data 4).

Additionally, certain non-m7G tRNAs also exhibited downregulation in senescent cells (Fig. 4E). Multiple mechanisms are known to regulate tRNA levels, such as the activity of RNA polymerase III, DNA methylation, and histone methylation at tRNA promoters. For example, certain genomic loci of transfer RNA (tRNA) showed age-dependent hypermethylation[46]. Studies have shown that p53 and RB1 act to suppress tRNA transcription, while c-Myc promotes tRNA transcription[47]. Hence, it is expected that METTL1 is involved in the reduction of specific tRNA levels (m7G-tRNAs), alongside other pathways that participate in the regulation of either specific tRNAs or all tRNA levels during senescence. Hence, various regulatory mechanisms contribute to a partial match between Fig. 4E, F.

Based on the abundance of m7G (Fig. 4D) and the transcript levels (Fig. 4E, G), we overlapped tRNAs in the above lists to identify thirteen m7G-tRNAs as potential candidates for METTL1-regulated and senescence-related tRNAs. The m7G-tRNAs were downregulated in METTL1 knockout-induced senescence at both the m7G methylation level and transcript levels (Fig. 4H). To validate the high-throughput quantification of tRNA levels, we selected five tRNAs out of thirteen to perform northern blotting. The study revealed significant decreases in Val-CAC, Val-AAC, Val-TAC, Pro-CGG, and Cys-GCA in senescent cells and METTL1 KO cells. However, Thr-AGT, a non-m7G tRNA, showed a slight increase in senescent cells (Fig. 4I). This result aligns with the sequencing data. Based on these results, we hypothesized that these thirteen tRNAs may function as effectors in METTL1-mediated senescence. We then explored how m7G-tRNA downregulation is linked to senescence regulation.

### Ribo-Seq uncovers ribosome stalling in METTL1-deficient cells

tRNAs function as adaptors, carrying amino acids to the mRNA-ribosome complex for mRNA translation. A decline in tRNA abundance may affect global protein synthesis[48–51]. Firstly, we conducted a puromycin incorporation assay to assess protein synthesis status during senescence. It was observed that puromycin incorporation was significantly reduced in senescent cells and METTL1-deficient cells, respectively (Fig. 5A). Ectopic expression of METTL1, not the AFPA mutant, reversed the declining trends of protein synthesis in senescent cells (Fig. 5A). Polysome profiling, commonly used to evaluate mRNA translation status, indicated lower polysome fractions in senescent cells and METTL1 knockdown (KD) cells compared to proliferative cells and scramble-transduced cells, respectively (Fig. 5B).

The above results suggest that global mRNA translation was hindered during METTL1-mediated senescence. However, it was unknown which mRNAs' translation was affected by METTL1 knockout. To address the question, we evaluated genome-wide mRNA translation using Ribo-seq to reveal codon usage and mRNA translation efficiency

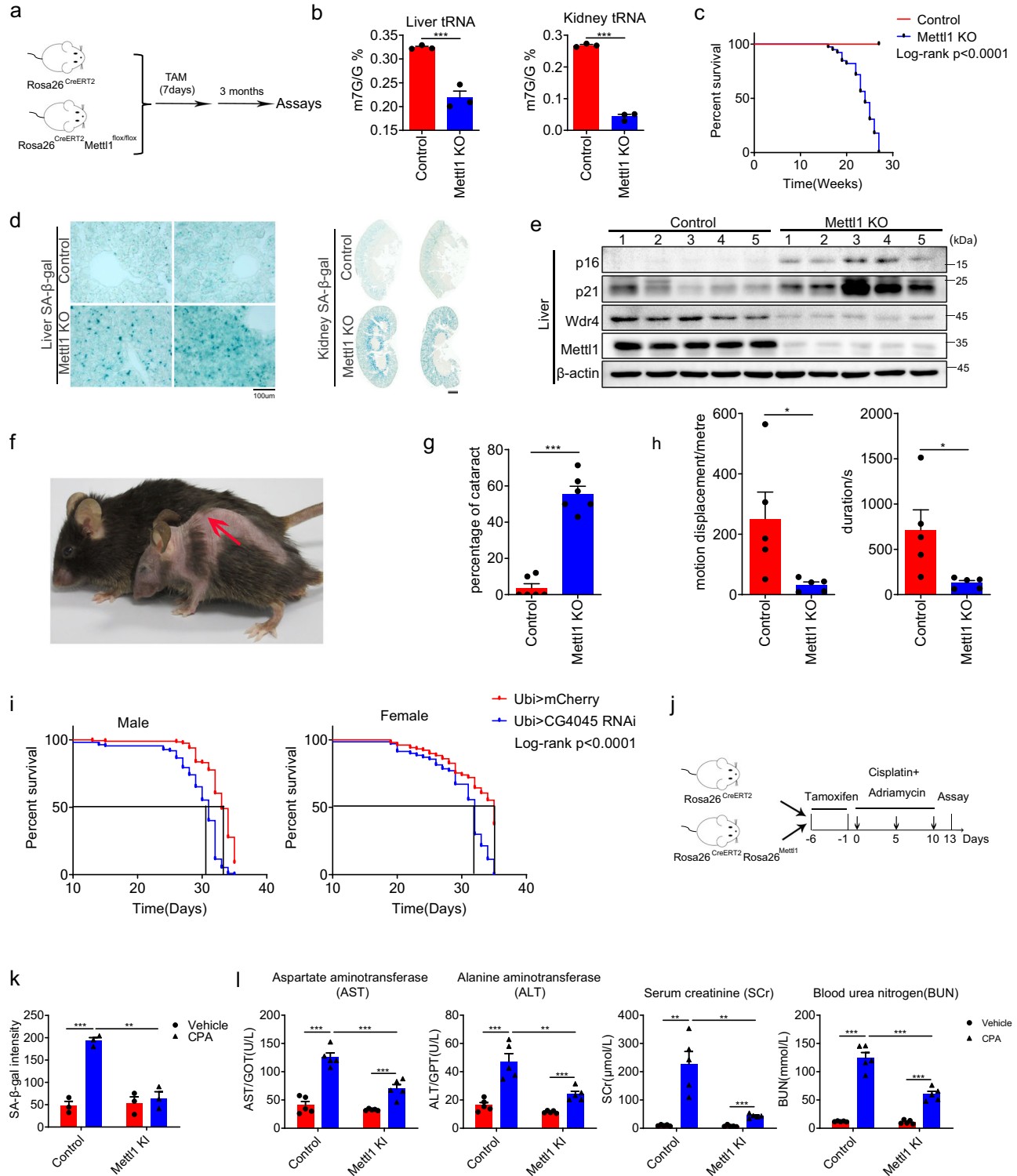

(TE)[30,52,53] (Fig. S5A−C). Analysis of Ribo-seq data for METTL1 knockout (KO) cells revealed that ribosomes were more likely to stall at m7G-tRNA-decoded codons compared to other codons (Fig. 5C, D). However, no significant difference was observed at the A + 1 site, suggesting a reduced translation of mRNAs decoded by m7G-tRNA (Fig. S5D, E). We conducted Gene Set Enrichment Analysis (GSEA) on mRNAs showing reduced translation efficiency (TE). The ribosome-related pathway, aging-associated degenerative diseases, p53 signaling pathway, WNT pathway, and Fanconi anemia pathway were enriched upon METTL1 depletion (Fig. 5E, Supplementary Data 5). These enriched

signaling pathways were closely associated with the observed senescence phenotypes. For example, WNT2, a ligand for the WNT pathway, has been identified as a significant regulator of senescence[54].

We validated the translation efficiency data from Ribo-Seq by quantifying the ratios of actively translating mRNAs to the total mRNAs isolated from ribosome-nascent chain complex-associated mRNAs (RNC-RNA)[55,56]. The data exhibited a decrease in the translation ratio of Wnt pathway-related proteins, including WNT2, CCND3, CTBP1, FZD2, MAPK8, and LEF1, following the knockout of METTL1 (Fig. S5F). WNT2 transcripts remained unchanged in both METTL1 knockout (KO) cells

**Fig. 3 | Depletion of Mettl1 induces premature aging. a** Schematic of the experiment for conditional knockout of Mettl1.**b** m7G modifications of tRNA from Mettl1 KO mice(2 female and 1 male) and control mice(2 female and 1 male) were detected using HPLC/MS. **c** Kaplan-Meier curves were generated for Mettl1 knockout (KO) and control mice. Control group: $n = 44$(24 female and 20 male); cKO group: $n = 39$(22 female and 17male mice). $P < 0.0001$. **d** Senescent cells in Mettl1 knockout (KO) liver and kidney tissues was assessed using SA-β-gal staining, one female and one male mouse in each group were used in the staining. Scale bar = 100 μm. **e** p16 and p21 were detected using Western blotting in liver tissue from control and knockout (KO) mice, five mice were used to isolate protein ($n = 5,3$ female and 2 male). **f** Representative photos of Mettl1 knockout and control mice around 6 months old were shown. The arrow points to postural kyphosis. **g** The percentage of mice with cataracts was calculated. Six biologically repeats were conducted, $n = 35$(20 female and 15 male). **h** Muscle performance was tested using a

treadmill exhaustion assay in control and Mettl1 knockout (KO) mice ($n = 5$ for each group, 3 female and 2 male). **i** The survival assay results of *Drosophila* depleted of CG4045 using UAS-GAL4 were presented. (male = 315; female = 278). **j** Schematic diagram for cisplatin and adriamycin (CPA)-induced chemotherapy senescence. **k** Liver sections from the CPA model were subjected to SA-β-gal staining, $n = 3$(2 female and 1 male). **l** Aspartate aminotransferase (AST), alanine aminotransferase (ALT), serum creatinine (SCr), and urea nitrogen (BUN) levels were measured for the CPA model, 5 male mice per group were used(3 female and 2 male). Unless otherwise indicated, all the assays were biologically repeated for three times, all data were presented as the mean ± SEM. $P$ values were calculated by two-tailed unpaired $t$ test (**b, g, h**), two-way ANOVA with Bonferroni's multiple comparisons test(k,l). *$p < 0.05$; **$p < 0.01$;***$p < 0.001$. Source data are provided as a Source Data file.

and senescent cells (Fig. 5F). Nonetheless, the proportion of actively translated WNT2 mRNA significantly decreased in METTL1 KO cells and in senescent cells as measured by QPCR with RNC-mRNA (Fig. 5G). A notable reduction in the protein levels of WNT2 and its downstream protein β-catenin was consistently observed in these cells (Fig. 5H, I). The findings suggest that the absence of METTL1 inhibited the Wnt signaling pathway.

To investigate if restoring Wnt signaling delays senescence, we provided METTL1-depleted cells with Wnt3a-conditioned medium. The study showed partial reversal of senescence phenotypes, as seen in SA-β-gal staining (Fig. S5G) and protein levels of EZH2 and LMNB1 (Fig. 5I). The data indicated that the Wnt signaling pathway played a role in METTL1-mediated senescence.

The aforementioned findings indicated a decrease in protein translation during senescence, corroborating earlier studies[6–8]. Rapamycin, an mTORC1 inhibitor, delays senescence and aging by enhancing autophagy and affecting protein synthesis in the short term[57,58]. We tried to investigate potential relationship between rapamycin and METTL1 in relation to senescence. The treatment of rapamycin to cells overexpressing METTL1 did not significantly change their senescence status, as assessed by the levels of senescence-associated proteins p16$^{INK4A}$ and p21$^{CIP1}$ (Fig. S5H). Therefore, the protein synthesis regulated by METTL1 aligns with the effect of rapamycin on lifespan.

After observing reduced mRNA-m7G levels in METTL1-deficient cells using HPLC-MS, we performed m7G-MeRIP to identify mRNAs with altered m7G methylation in these cells. By analyzing the data, we have found overlapping mRNAs with decreased internal m7G methylation levels, as indicated by MeRIP, along with their translation efficiency (TE) determined through Ribo-Seq analysis (Fig. 5E). It is evident that these common mRNAs are notably enriched in the WNT pathway, ribosome, and neurodegenerative diseases (Fig. 5J). Thus, the involvement of mRNA-m7G in senescence due to METTL1 deficiency could be plausible to some extent.

## METTL1 downregulation elicits SASP through ribotoxic stress response and integrated stress response

Previous studies indicated that persistent ribosome stalling causes ribosome collisions, leading to the formation of di-ribosomes[59–67]. Colliding ribosomes recruit E3 ubiquitin ligases Hel2 (ZNF598) and RACK1 to separately ubiquitinate 40S ribosome proteins RPS10/RPS20 and RPS2/RPS3. This process is essential for initiating ribosome quality control[64,66,68–74]. To confirm ribosome stalling, we measured the ubiquitination of ribosomal proteins by isolating ribosomes with a 1 M sucrose cushion. Western blotting of isolated ribosomes showed elevated levels of ubiquitylated RPS20 protein in senescent and METTL1-depleted cells (Fig. 6A). The data indicated that ribosome quality control (RQC) response was activated in senescent cells and METTL1 knockout cells. We evaluated if stress response signaling pathways, such as the integrated stress response (ISR) and the SAPK p38-JNK signaling pathway (Ribotoxic stress response), typically associated

with ribosome quality control[61,66,75,76], were activated in senescent cells and METTL1 knockout (KO) cells. Our findings showed a significant upregulation of phosphorylated eIF2α, p38, JNK, and ATF4 proteins in senescent cells and aged mouse tissues, downstream of ISR and RSR (Fig. 6B, C and Fig. S6A). Additionally, two stress responses were activated in METTL1-depleted IMR90 cells and Mettl1 KO mouse liver and kidney tissues (Fig. 6D, E and Fig. S6B).

Overexpressing intact METTL1, not the AFPA mutant, in senescent cells alleviated the integrated stress response (ISR) and RSR response, as shown by decreased levels of phosphorylated eIF2α, p38, and JNK proteins (Fig. 6F). The results demonstrated that METTL1 reduction during senescence and aging significantly activated the integrated stress response (ISR) and RSR.

There are four known kinases that phosphorylate eIF2α to inhibit global protein translation initiation[77,78]. Since HRI is not present in IMR90 cells, we used chemical inhibitors for PKR, PERK, and GCN2 to identify the kinase responsible for eIF2α phosphorylation. The results showed that only the GCN2 inhibitor, A-92, reduced p-eIF2α to the basal level (Fig. 6G).

ZAKα was reported to be activated and necessary for the early activation of SAPK (p38/JNK) following ribosome stalling[61,67]. Therefore, we investigated if ZAKα is involved in activating SAPK (p38 and JNK) due to METTL1 deficiency. After interfering with ZAKα using siRNA, the phosphorylation of p38, JNK, and eIF2α was inhibited in METTL1-depleted cells (Fig. 6H). Persistent stalled ribosomes activated the Integrated Stress Response (GCN2-ISR) and the p38/JNK signaling cascade by activating ZAKα. Given p38's role as a key regulator of SASP[79], it is hypothesized that ribotoxic stress responses may contribute to SASP activation in this context. Several senescence-associated secretory phenotype (SASP) factors, such as IL-6, CXCL8, and IL-1β, were upregulated in IMR90 cells when METTL1 was depleted, as shown in RT-qPCR results (Fig. 6I). We utilized chemical inhibitors A-92 (for GCN2), SB203580 (for p38), and SP600125 (for JNK) to assess their effect on SASP. These inhibitors can fully block the activation of their target (Fig. S6C). It was demonstrated that a combination of three inhibitors (3i, A-92 + SB203580 + SP600125) effectively suppressed the expression of senescence-associated secretory phenotype (SASP) factors in cultured METTL1 knockdown (KD) IMR90 cells. Individual inhibitors A-92 or SB203580 exhibited a similar effect to three inhibitors on specific SASP factors (Fig. 6I). Moreover, senescence-associated secretory phenotype (SASP) cytokines, including IL-6, were induced in Mettl1-deficient mice, as shown by ELISA assay (Fig. 6J). Administration of two inhibitors (A-92 and SB203580) to Mettl1 knockout (KO) mice reduced IL-6 levels (Fig. 6J).

Furthermore, the treatment of METTL1 knockdown cells with inhibitors A-92, SB203580, and SP600125 significantly decreased the number of SA-β-gal positive cells (Fig. 6K). These results suggested that METTL1 deficiency caused ribosome stalling, activated ISR and RSR responses, and increased the expression of SASP-related genes.

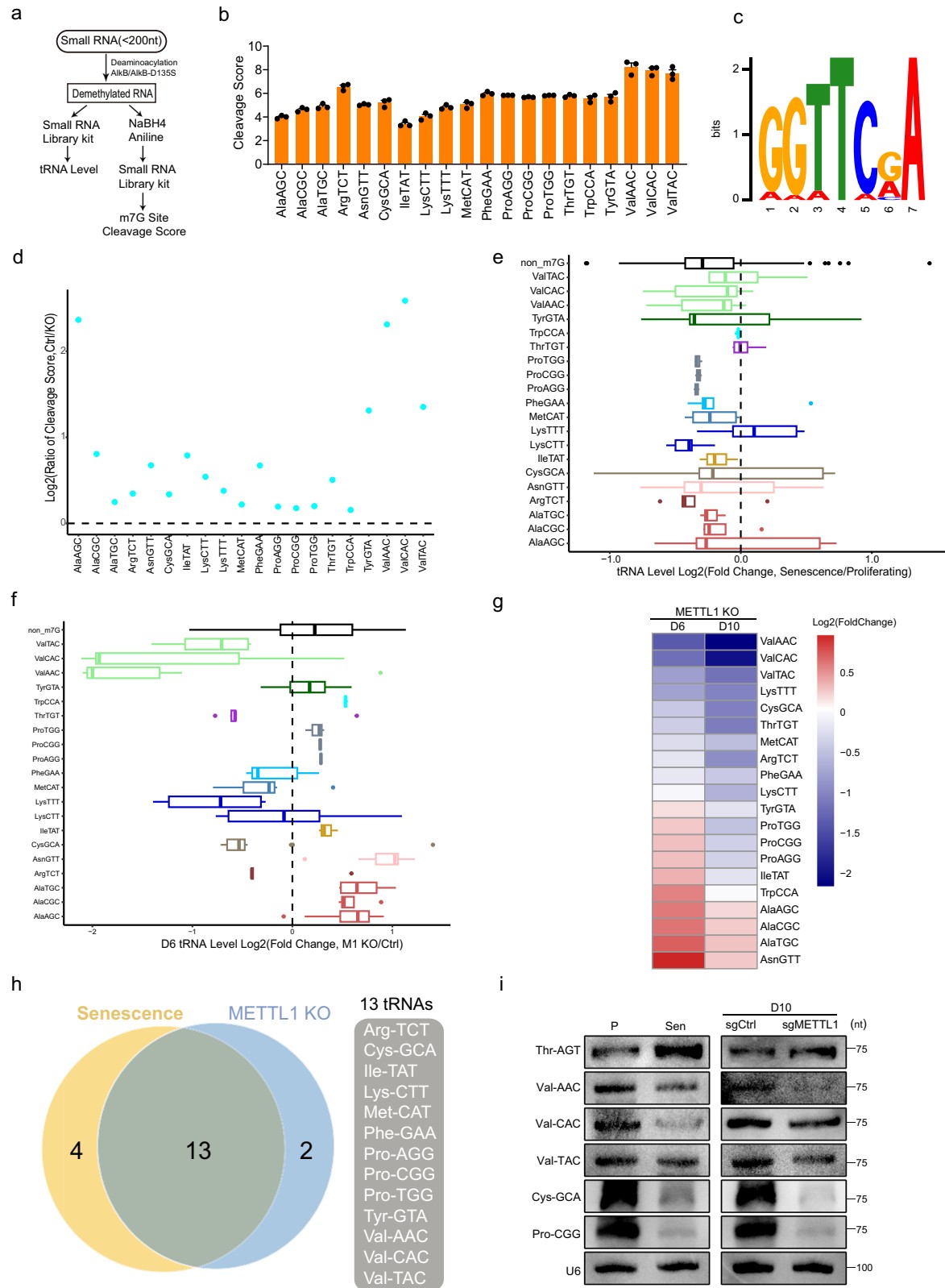

Restoring METTL1 expression or using ISR inhibitors can reduce SASP senescence phenotypes.

**RTD is involved in the senescence caused by METTL1 deficiency**
The data showed that decreased tRNAs due to METTL1 deficiency dampened global protein synthesis, leading to ISR and RSR stress responses. Hypomethylated m7G-tRNAs were thought to be degraded

quickly through the RTD pathway[33,80,81]. Previous studies have shown that the translation elongation factor eEF1A, responsible for delivering tRNA to the ribosomal A site, can bind to m7G-hypomethylated tRNAs and sequester them from RTD degradation[80]. The protein level of eEF1A was significantly downregulated in replicative senescence, ETO-induced senescence, and METTL1 deficiency-induced senescence in a time-dependent manner (Fig. 7A). The data suggested that tRNA

**Fig. 4 | METTL1 deficiency downregulates a group of m7G-tRNAs. a** Schematic diagram for TRAC-Seq experiment used to define m7G modification and assess tRNA levels. **b** Cleavage scores calculated from TRAC-Seq for tRNAs in IMR90 cells were exhibited. TRAC-Seq data were collected from three independent experiments. **c** Motifs for m7G methylation sites were summarized from TRAC-Seq data. **d** Cleavage scores ratios in sgRNA scramble control and METTL1 knockout (KO) cells were characterized. Cleavage Score Ratio = Cleavage Score of Control/Cleavage Score of Knockout. **e** Differentially expressed m7G-tRNAs were observed in proliferating and senescent cells. Each boxplot represents the expression of a tRNA type that was calculated from the combined expression of all the tRNA genes belonging to the same tRNA type. **f** Differentially expressed m7G-tRNAs in control/

METTL1 KO (Day 6) cells were observed. All tRNA genes of the same tRNA type were combined to calculate their transcript abundance. **g** Fold change in m7G-tRNA transcript abundance was presented for METTL1 KO compared to Ctrl. D6 and D10 mean 6 and 10 days after sgRNAs transduction, respectively. **h** Venn diagram showed the thirteen tRNAs regulated by METTL1 and senescence. The list was based on alterations in tRNA levels and m7G abundance in proliferating/senescent IMR90 cells and control/METTL1 KO cells. **i** Northern blotting was performed for tRNAs using the specified samples. The representative experiment was shown, three repeats demonstrate similar results. Source data are provided as a Source Data file.

stability and delivery were compromised in senescent cells because of eEF1A downregulation. Thus, we investigated whether eEF1A regulates METTL1-mediated senescence by introducing an ectopic eEF1A ORF-containing lentiviral vector into METTL1 knockout cells. The results showed that reintroducing the eEF1A protein in METTL1 knockout (KO) cells enhanced cell growth by two folds and reduced the number of SA-β-gal staining cells to 30% compared to the control group (METTL1 KO without eEF1A overexpression) (Fig. 7B). Moreover, the transduction of eEF1A in METTL1-deficient cells also reversed senescence-associated proteins, including p16$^{INK4A}$, p21$^{CIP1}$, LMNB1, and EZH2 (Fig. 7C). The levels of m7G-tRNAs and the overall protein synthesis capacity were partially restored in these cells, as demonstrated by Northern blotting and puromycin incorporation assay, respectively (Fig. 7D, E). Moreover, occurrences of ribosome stalling, as evidenced by the ubiquitination of RPS20, were partially reduced following the introduction of eEF1A in METTL1 knocked-out cells (Fig. 7F). Consequently, the ribotoxic stress response and integrated stress response exhibited a significant reduction, as indicated by the decreased phosphorylation levels of eIF2α, p38, JNK, and ATF4 proteins (Fig. 7G). The evidence demonstrated the involvement of eEF1A in METTL1-mediated senescence. The eEF1A protein decreased during senescence. We further examined the regulatory mechanisms governing this process. The eEF1A transcript remained unchanged (Fig. S7A); however, the translation efficiency of eEF1A, as measured by RNC-mRNA qPCR, was significantly reduced in the absence of METTL1 (Fig. 7H). Overexpression of METTL1, not AFPA mutant, partially reversed the decline in eEF1A levels during senescence (Fig. 7I). This suggests that eEF1A translation is regulated by m7G modification. Our findings illustrated that METTL1 deficiency caused ribosome stalling by reducing m7G-tRNAs, inhibiting protein translation initiation and elongation through eIF2α phosphorylation, and decreasing eEF1A levels.

Stalled ribosomes are usually resolved by ribosome quality control (RQC). If the system is overwhelmed, nascent peptides may aggregate[73]. In this study, we utilized the FlucDM-GFP[82], a proteostasis reporter, to examine whether protein aggregation occurs in METTL1-induced senescence. The FlucDM-GFP reporter exhibited elevated protein aggregation in senescent IMR90 cells, as determined by quantifying the number of GFP-aggregated inclusions. However, overexpression of METTTL1 reduced the aggregation in a m7G activity-dependent manner (Fig. 7J, Fig. S7B). Furthermore, the knockout of METTL1 in the proliferating cells increased FlucDM-GFP aggregation compared to control cells, but introducing eEF1A reversed this effect (Fig. 7K). These results suggest that the RNA m7G modification by METTL1 is essential for maintaining efficient protein translation and proteostasis.

### SP1 regulates METTL1 and WDR4 expression
The downregulation of METTL1 has been elucidated to induce senescence through the regulation of protein translation. However, the regulation of the METTL1 transcript remains unknown. To address the question, we examined the potential binding of transcription factors throughout the promoter region using PROMO[83]. The analysis

identified several transcription factors, including p53, E2F family proteins, c-Myc, YY1, WT1, and SP1. Given that p53 and c-Myc play crucial roles in the regulation of senescence, we have opted to conduct further investigations on these factors. Nevertheless, the overexpression of p53 or the knockdown of c-Myc did not have an impact on METTL1 transcripts (Fig. S8A). It has been documented that SP1 could regulate METTL1 expression in cancer cells[84]. Hence, there is a question regarding whether SP1 also regulates the expression of METTL1 during senescence. The expression level of SP1 was notably reduced during senescence in IMR90 cells (Fig. 8A). Subsequently, short hairpin RNAs (shRNAs) were employed to knock down SP1 in IMR90 cells. Upon SP1 knockdown, the transcription levels of both METTL1 and WDR4 exhibited a reduction of approximately 50% (Fig. 8B, left panel), Moreover, restoring SP1 in senescent cells slightly rescued METTL1 and WDR4 expression (Fig. 8B, right panel). ChIP-PCR results indicated direct binding of SP1 to METTL1 promoter regions around TSS sites (Fig. 8C). Furthermore, we employed a dual luciferase reporter assay system to confirm the regulatory impact of SP1 (Fig. S8B). The 1k base pairs of the METTL1 promoter exhibited a significant ability to drive luciferase expression, however, when the binding sites on the promoter were mutated, its activity was significantly lost (Fig. 8D). Upon co-transfection of ectopic SP1 into cells, there was a further enhancement in the luciferase activity driven by the promoter (Fig. 8D). The binding of SP1 was hindered during senescence due to reduced SP1 levels (Fig. 8E). The findings showed that the transcription of METTL1 and WDR4 SP1 is controlled by SP1 during senescence.

Considering the demonstrated influence of SP1 on METTL1 transcription, our investigation focused on determining if SP1 also regulated the cellular protein synthesis activity. Consistently, the inhibition of SP1 led to a reduction in protein synthesis activity (Fig. 8F). Furthermore, SP1 was found to regulate senescence, as shown by EdU assay and SA-β-Gal staining (Fig. 8G, H). Indeed, a previous study has demonstrated that SP1 plays a role in senescence through the regulation of nucleocytoplasmic trafficking (NCT)-related gene expression[85]. We found that the decrease in SP1 levels during senescence could result in the downregulation of METTL1 and WDR4 expression, thereby disrupting protein translation and ultimately leading to senescence. This discovery revealed an additional role of SP1 in the process of senescence.

Taken together, our study elucidated the role of RNA m7G modification in mediating protein synthesis and proteostasis, thereby regulating senescence. Downregulation of SP1 hindered the transcription of METTL1 and WDR4. METTL1 deficiency decreased RNA m7G methylation. Hypomethylated transfer RNAs (tRNAs) at the N7-methylguanosine (m7G) position were directed towards the rapid tRNA decay (RTD) pathway, resulting in ribosome stalling at specific mRNA codons. Persistent ribosome stalling triggers the ribotoxic stress response and integrated stress response, leading to the activation of SASP and senescence phenotypes. The decrease in translation efficiency of senescence-associated mRNAs, such as WNT2, also contributes to senescence phenotypes. Restoration of eEF1A expression reversed senescence phenotypes induced by METTL1 deficiency through alleviating rapid tRNA degradation (RTD) (Fig. 8I).

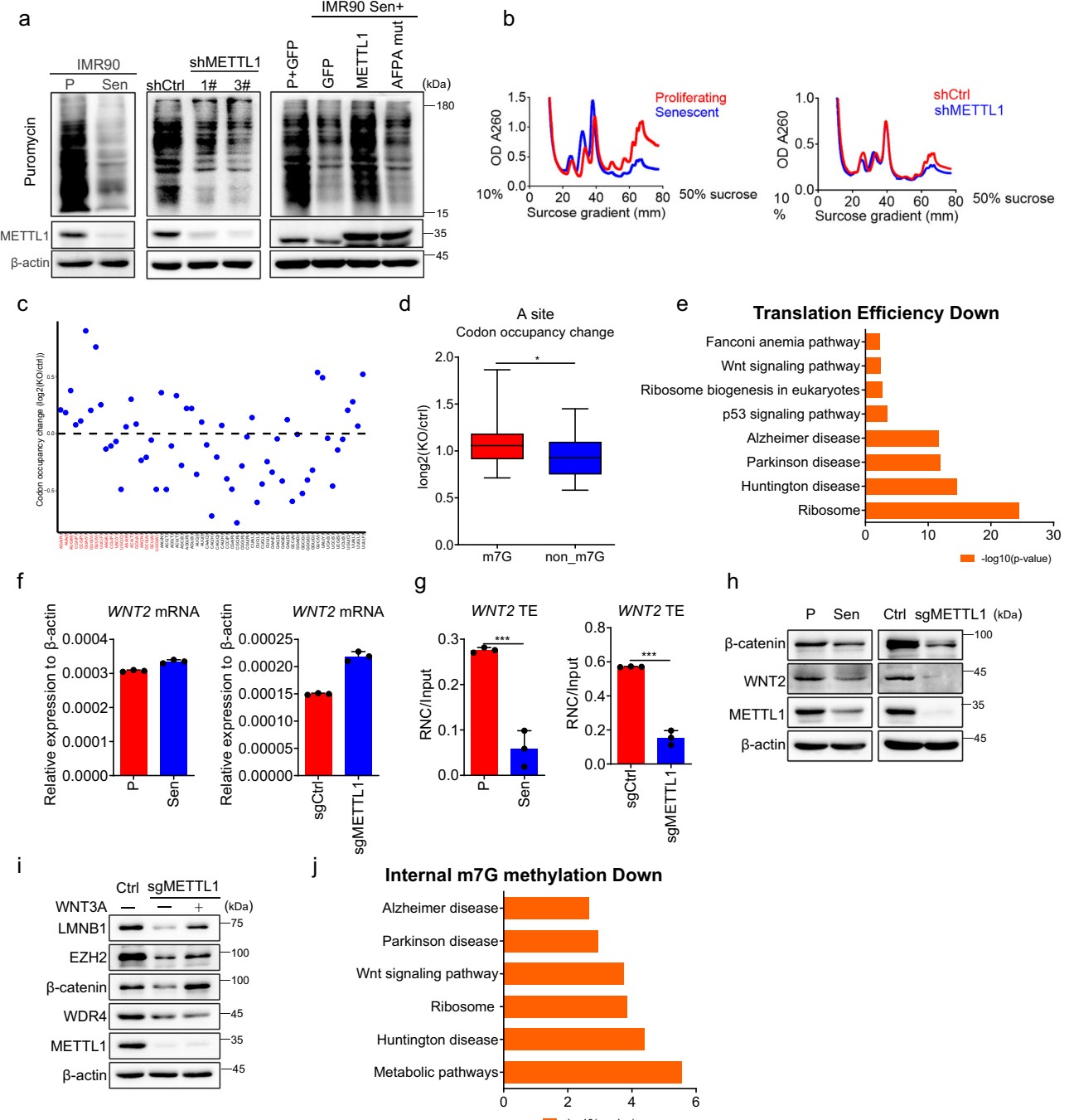

**Fig. 5 | Ribo-seq uncovers ribosome stalling in METTL1-deficient cells.**
**a** Puromycin incorporation assay was performed in young and senescent cells, scramble control and METTL1 knockdown (KD) cells, and cells with ectopic expression of GFP, METTL1, and AFPA mutant. **b** Polysome profiling was conducted for senescent cells, METTL1 KD cells, and their corresponding control cells. **c** Ribosome occupancy at individual codons in the A site. Plots represent the relative ribosome footprinting signals from sgRNA and scramble control cells. Two replicates were performed. The codons are separated into m7G-tRNA decoding codons(red) and non-m7G-tRNA decoding codons(black). **d** Mann–Whitney *U* test on the ribosome occupancy at individual codons in A site from m7G-tRNA and non-m7G-tRNA. The ribosome footprinting signals from sgRNA and control samples were analyzed. The boxplot showing the mean, standard deviation (std), and median for each tRNA type. The upper and lower limits and the center line represent the 75th and 25th percentiles and the median, respectively; upper and lower whiskers indicate ±1.5× the interquartile range. *$p$ < 0.05. **e** Bar plots show the

enriched KEGG terms of decreased TE genes detected by Ribo-Seq upon METTL1 knockout. Two-sided Fisher's exact test was used without adjusting for multiple comparisons in this analysis. **f** mRNA levels of WNT2 were detected in senescent IMR90 cells and METTL1-deficient cells using RT-qPCR. **g** The translation efficiency of WNT2 for senescent IMR90 cells and METTL1 KO cells was measured by RT-qPCR using RNC-mRNAs. **h** WNT2 protein was detected by Western blotting in METTL1 KO IMR90 cells and senescent IMR90 cells. **i** Senescence-associated markers were detected by Western blotting in METTL1 KO IMR90 cells with or without Wnt3a-conditioned medium. **j** Bar plots show the enriched KEGG terms of genes with reduced internal m7G methylation levels, as detected by MeRIP, and decreased translation efficiency (TE). KEGG analysis for the shared mRNAs are presented. The representative result was presented, three repeats demonstrate similar results (**a**, **b**, **h**, **i**); Three repeats were performed for qPCR assay. **f**, **g**. All data were presented as the mean ± SEM. Two-tailed unpaired *t* test (**f**, **g**). *$p$ < 0.05, **$p$ < 0.01, ***$p$ < 0.001. Source data are provided as a Source Data file.

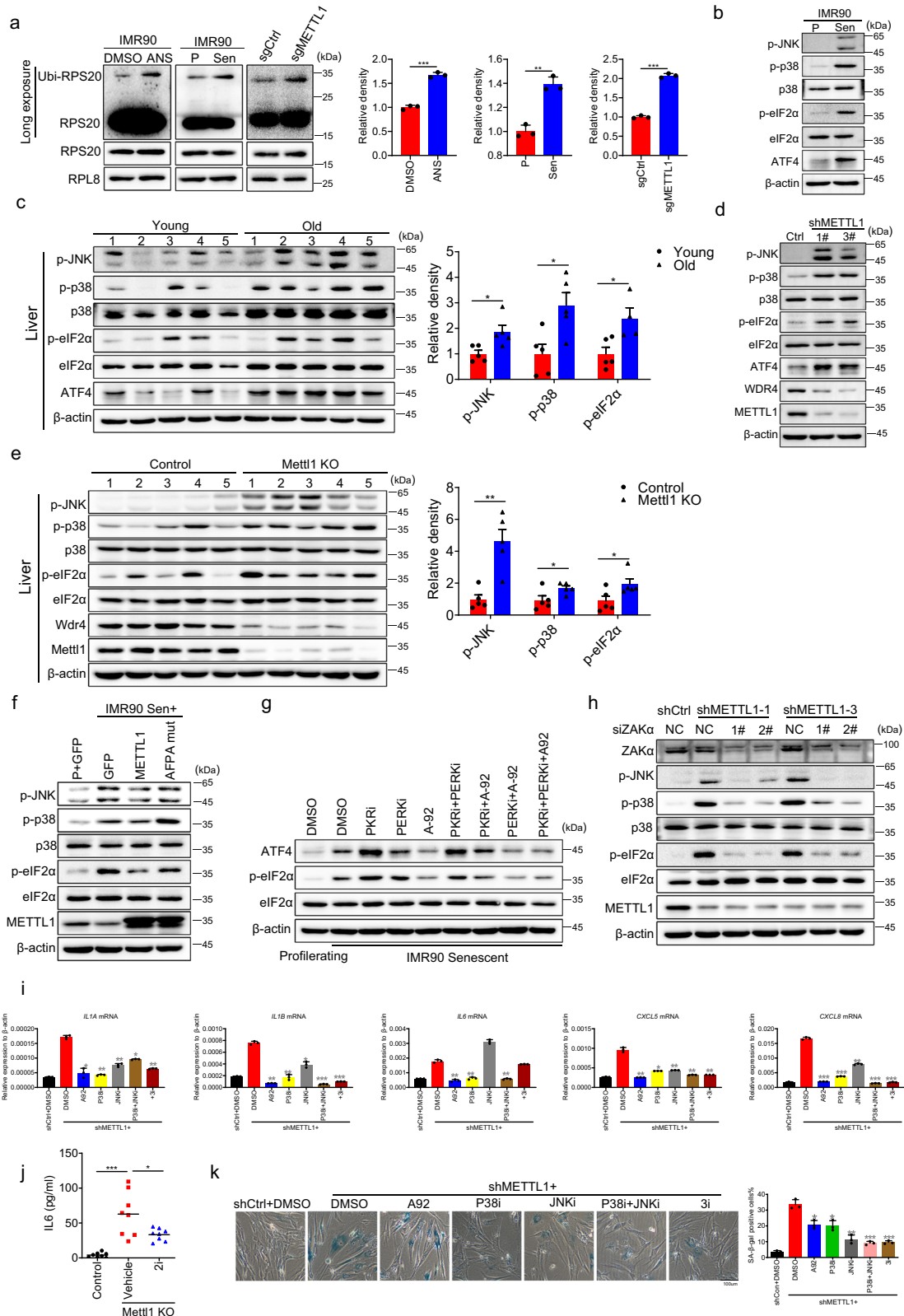

## Discussion

In the study, we demonstrated the essential role of METTL1-mediated tRNA m7G modification in protein synthesis and its relevance in senescence and ageing. As previously reported, a group of RNAs was recognized as substrates for METTL1[43,86–89]. PAR-CLIP revealed that 31.8% of METTL1 binding RNAs belong to tRNAs, which are the most abundant RNA species in the assay[90]. Our experiments focused on

tRNAs to explore the mechanisms of METTL1 mediated senescence. However, we cannot exclude the possibility that other types of RNAs with m7G modification also contributed to METTL1-induced senescence and aging phenotypes. Further studies in this direction can help to address the issue.

It is largely unknown how m7G46 modification regulates tRNA stability. However, relevant studies in yeast provided some clues. It was

**Fig. 6 | METTL1 deficiency elicits SASP through ribotoxic stress response and integrated stress response. a** Ubiquitination of RPS20 in isolated ribosomes from indicated samples was detected using western blotting. ANS-treated IMR90 cells served as the positive control. Arrow-pointed bands indicated ubiquitination of the RPS20 protein. **b** ISR and RSR response-relevant proteins, were detected by western blotting in proliferating/senescent IMR90 cells(**b**), young/aged liver tissues, *n* = 5 mice per group(2 female and 3 male) (**c**), METTL1 KD/control cells(**d**), liver tissues from Mettl1 KO and control, *n* = 5 mice per group(3 female and 2 male) (**e**), and METTL1 and mutated AFPA mutant transduced IMR90 cells (**f**) were detected using western blotting. **g** Phosphorylated eIF2α and ATF4 proteins were detected in senescent cells treated with various inhibitors using western blotting. **h** Ribotoxic stress response proteins were examined upon the inhibition of ZAKα by delivering siRNAs into METTL1 KD cells. **i** SASP factors were measured by RT-qPCR in METTL1 knockdown cells with or without inhibitors, as indicated. **j** The IL-6 levels in Mettl1 knockout (KO) mice with or without inhibitor and control mice were detected using ELISA (*n* = 8 per group, 4 female and 4 male). **k** The impact of GCN2, p38, and JNK inhibitors on Mettl1-mediated senescence status was assessed through SA-β-gal staining. Scale bar = 100 μm. All in vitro assays were biologically repeated three times. All WB results were performed three times, the representative result was shown, the protein intensity was quantified with Image J (**a**, **c**, **e**); All data were presented as the mean ± SEM. Two-tailed unpaired t test (**a**), one-way ANOVA with Bonferroni's multiple comparisons test (**c**, **e**, **i**–**k**). *$p < 0.05$, **$p < 0.01$, ***$p < 0.001$. Source data are provided as a Source Data file.

observed that tRNAs in Saccharomyces cerevisiae with trm8Δtrm4Δ genotype that was deficient in m7G modification were degraded through rapid tRNA decay (RTD) pathway which was mediated by Rat1, Xrn1 and Met22[33,81]. Depletion of both Rat1 and Xrn1 or Met22 alone can restore tRNA-Val-AAC level and rescue trm8Δ phenotypes[81]. Additionally, another two proteins, TEF1 and VAS1, which separately encodes elongation factor eEF1A and valyl-tRNA synthetase, also restore tRNA-Val-AAC level by competing with RTD to sequestered hypomethylated tRNA from degradation[80,91]. Our results indeed proven the point that restoration of eEF1A suppressed tRNA decay and partially reversed METTL1 deficiency-mediated senescence.

In addition to tRNA stability, transcription regulation also participates in the maintenance of tRNA homeostasis. Indeed, our results showed that downregulated tRNAs in senescent cells included both m7G-containing tRNAs and non-m7G tRNAs. The data suggest that transcription step may also participate in the regulation of tRNA levels during senescence in addition to tRNA stability which is related to m7G methylation.

A recent report indicated that ribosome collisions occur with ageing[92]. Our study found that METTL1 deficiency-mediated hypomethylation of m7G in tRNAs leads to ribosome pausing and stalling during senescence. Whether ribosome collisions occur in this context is worth further exploration.

It is demonstrated that mTORC1 activity is enhanced during senescence[93–96]. mTORC1 promotes cell growth by regulating protein synthesis and lipid metabolism and suppressing protein turnover(autophagy)[97]. In response to various environmental factors, mTORC1 promotes protein synthesis mainly through two key effectors: p70S6 Kinase 1 (S6K1) and eIF4E Binding Protein (4EBP), which facilitate mRNA translation initiation[97]. Accordingly, senescent cells are theoretically believed to have upregulated protein synthesis and suppressed autophagy. Indeed, many studies have shown that autophagy activity declines with aging[98]. Restoration of protein turnover by inhibiting mTORC1 can extend lifespan[99]. However, experimental data from various organisms and tissues have shown that protein synthesis decreases during the aging stage[6–8]. This inconsistency may stem from the complex regulation of mRNA translation, which includes factors such as tRNA and amino acid availability, ribosome machinery assembly, and other unidentified factors. This study demonstrates that METTL1-mediated tRNA stability plays a crucial role in maintaining global proteome homeostasis, which is essential for preventing premature senescence. These results underscored the important physiological roles of m7G modification in the regulation of aging. The discovery may offer a new potential intervention for healthy aging.

## Methods
### Mice experiments
Animal experiments were carried out following protocols approved by the Institutional Animal Care and Use Committee (IACUC) for animal research at the Guangzhou Institutes of Biomedicine and Health, CAS. Mice in the study were kept in constant temperature (25 °C) and humidity (60%) conditions with a 12: 12 hr light: dark cycle. The Rosa26^CreERT2 mouse strain was obtained from the Shanghai Model Organisms Center. Mettl1-cKO and Mettl1-cKI mice were generously provided by Professor Shuibing Lin from the First Affiliated Hospital, Sun Yat-sen University. All mice in the study is C57BL/6J strain background and were housed in a specific pathogen-free (SPF) environment under standard conditions. Both male and female mice are utilized in all animal experiments. For the chemotherapy-induced senescence model, cisplatin and Adriamycin (CPA) were administered at doses of 9 mg/kg and 6 mg/kg to 8-12-week-old mice (both male and female) via intraperitoneal injection. Mice received injections every five days for a total of three administrations, after which blood samples were collected for biochemical assays. Corn oil containing tamoxifen at a concentration of 10 mg/ml was administered to conditional cKO or cKI mice via oral gavage for seven consecutive days to induce recombination. After a 3-month period following the last administration of tamoxifen, the conditional knockout (cKO) mice were subjected to aging assays. Mice aged between 20 and 24 months were categorized as the old group, while those aged 2 to 3 months were classified as the young group.

### Drosophila stocks and general husbandry
The standard laboratory stocks w1118 (VDRC60000) were used in our experiments. The CG4045 RNAi fly was obtained from the Tsinghua Fly Center (THFC #TH201500223.S). Ubi-Gal4, tub-Gal80ts flies were generated by Dominique Ferrandon team. Ubi-mCherry flies were used as control, generated by crossing Ubi-Gal4, tub-Gal80ts virgins to UAS-mCherry RNAi (Bloomington Stock Center #BL35785); Ubi-CG4045 RNAi flies were generated by crossing Ubi-Gal4, tub-Gal80ts virgins to UAS-RNAi-CG4045. For all experiments, larvae were cultured in corn-meal-sugar-yeast media. The crossing was performed under 25 °C for three days, after which the vials containing larvae were transferred to 18 °C until adults hatched. Same-age adults were collected within a 24-h window and transferred to 10% Sugar/Yeast (SY) food. Flies were transferred to fresh vials every two days.

**Survival assays.** Lifespans were measured using previously reported protocols[100]. Normally, 20 replicate vials (around 300 experimental flies) were established for each treatment. Flies were counted every day and transferred to fresh media every two days; Flies were kept in constant temperature (25 °C) and humidity (60%) conditions with a 12: 12 h light: dark cycle. For RNAi experiments, the flies were housed at 29 °C throughout their lifespan.

### Cell culture
IMR90 primary human diploid lung embryonic fibroblasts were purchased from ATCC (CCL-186) and cultured in DMEM medium with 10% fetal bovine serum (FBS) and 1% penicillin-streptomycin under 3% oxygen. Cells at confluence were split at a 1:4 ratio. 293 T (ATCC, CRL-3216) cells were cultured in DMEM medium with 10% FBS and 1% penicillin-streptomycin under 5% $CO_2$. All the senescence experiments were performed using IMR90 cell.

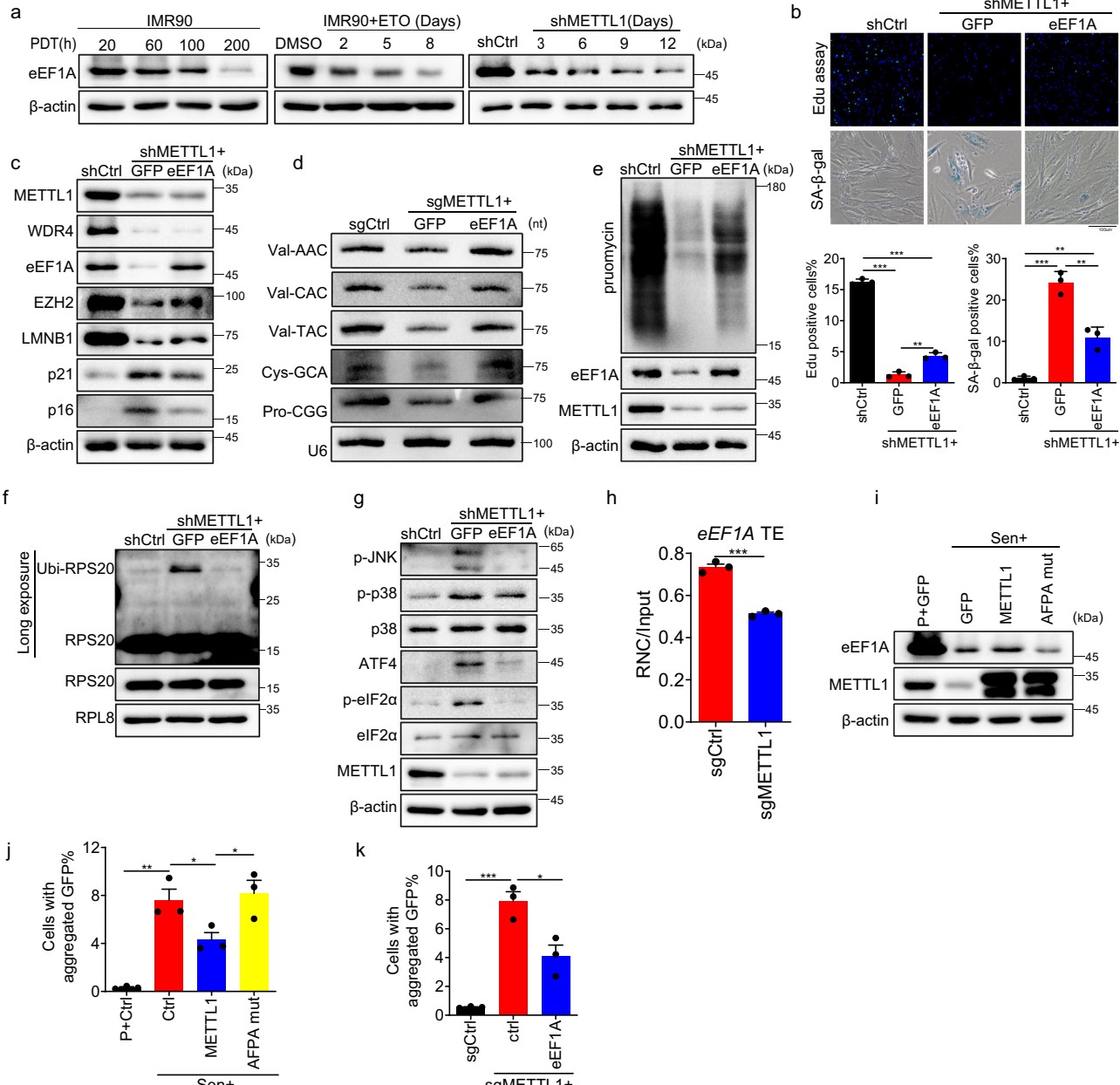

**Fig. 7 | RTD is involved in the senescence caused by METTL1 deficiency. a** The eEF1A protein level was detected by Western blotting using samples from replicative senescence, ETO-induced senescence, and METTL1-depleted IMR90 cells at various time points as indicated. **b** Cell proliferation and SA-β-Gal-positive senescent cells were analyzed following eEF1A expression in METTL1-depleted cells. **c** Senescence markers were analyzed following eEF1A expression in METTL1-depleted cells. **d** Northern blotting was conducted using METTL1 knockout cells with or without eEF1A overexpression to detect several tRNA level. **e** Puromycin incorporation was analyzed by Western blotting when eEF1A was expressed in METTL1-depleted cells. **f** Ubiquitination of RPS20 was assessed in isolated ribosomes from control cells, METTL1 knockout cells, with or without eEF1A overexpression, using a 1 M sucrose cushion, and detected by western blotting.Arrow-pointed bands were ubiquitinated RPS20 protein. **g** Phosphorylated eIF2α, JNK, p38, and ATF4 proteins were detected by Western blotting in METTL1-depleted

cells with or without eEF1A overexpression. **h** The active translation ratios of eEF1A in METTL1 KO and control cells were determined by qPCR using RNC-mRNA. **i** eEF1A level was detected in senescent cells overexpressing GFP control, METTL1 and AFPA mutant by Western blotting. The protein aggregation in senescent IMR90 cells with METTL1, AFPA-mutant, or an empty vector control(**j**) and METTL1 KO cells with or without eEF1a (**k**) was evaluated using the FlucDM-GFP reporter. The aggregated protein percentage is determined by the aggregated GFP inclusions cell against the total GFP-positive cells. The representative result was shown, three repeats demonstrate similar results (**a, c–g, i**). All the assays were biologically repeated three times. All data were presented as the mean ± SEM. Two-tailed unpaired *t* test (**h**), one-way ANOVA with Bonferroni's multiple comparisons test (**b, j, k**). *$p < 0.05$, **$p < 0.01$, ***$p < 0.001$. Source data are provided as a Source Data file.

## Proteomics assay

Early passage (p8) and late IMR90 cells (p26) were lysed using 8 M urea, 10 mM DTT, and 0.1 M pH 8.5 Tris buffer. Cells were sonicated for 10 cycles at 4 °C using a Diagenode Biorupter Pico Sonicator. The lysate was then centrifuged at maximum speed at room temperature

for 15 min. Whole proteome peptides were prepared using the FASP protocol as previously described[101]. Soluble lysates were added to a 10 kDa cutoff filter (MRCF0R010, Millipore) and centrifuged at 11500 *g* at 20 °C for 15 min. 50 mM iodoacetamide (IAA, Sigma, I1149) in urea buffer was used to alkylate proteins at 20°C for 15 min. After washing

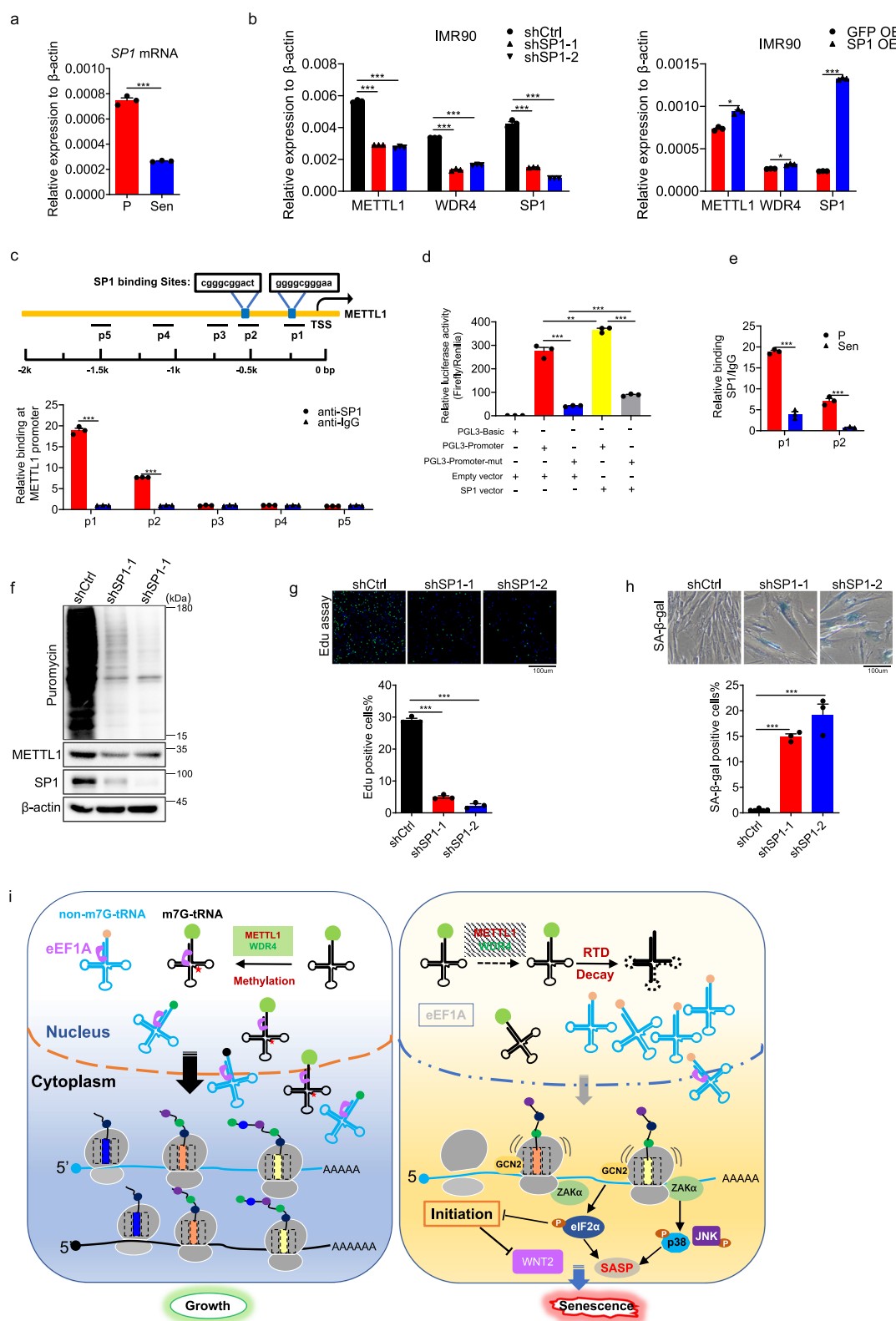

**Fig. 8 | SP1 regulates METTL1 and WDR4 expression. a** SP1 level was measured in proliferating cells and senescent cells. **b** Level of METTL1 and WDR4 were quantified in cells with SP1 knockdown or SP1 overexpression. **c** The presence of SP1 on the METTL1 promoter was evaluated using chromatin immunoprecipitation followed by polymerase chain reaction (ChIP-PCR). **d** METTL1 promoter activity was assessed by measuring luciferase activity and normalizing it with TK renilla activity. **e** The binding of SP1 on the METTL1 promoter was evaluated in proliferating and senescent cells using ChIP-PCR. **f** Protein synthesis was assessed using a puromycin incorporation assay in sp1 depleted cells. Representative result was shown. **g** Cell proliferating was evaluated by EdU assay in cells that were depleted of SP1. Scale bar = 100 μm. **h** Senescent cell upon SP1 depletion was evaluated by SA-β-gal staining. Scale bar = 100 μm. **i** Schematic diagram of model for METTL1 deficiency-induced cell senescence. All the assays were repeated three times. All data were presented as the mean ± SEM. Two-tailed unpaired *t* test (**a**, **e**), one-way ANOVA with Bonferroni's multiple comparisons test (**b**–**d**, **g**, **h**). *$p < 0.05$, **$p < 0.01$, ***$p < 0.001$. Source data are provided as a Source Data file.

with urea lysis buffer and 50 mM ammonium bicarbonate (ABC) buffer, the protein was digested with trypsin in ABC buffer overnight at 37°C in a wet chamber. Peptides were extracted using a 50 mM ABC buffer and the pH was adjusted to around 2 with 10% TFA. Desalted peptides were separated and analyzed using an Easy-nLC 1200 connected online to a Fusion Lumos or Fusion Eclipse mass spectrometer equipped with FAIMS pro. The analysis involved a gradient of buffer B (80% acetonitrile and 0.1% formic acid) with a total of 240 min: 1 min at 2%, 10 min from 2% to 7%, 200 min from 7% to 28%, 15 min from 28% to 36%, 5 min from 36% to 60%, and 7 min at 95% of buffer B. The raw data were initially analyzed using MaxQuant version 1.6.17.0 by searching against the Human Fasta database.

## Plasmid construction
sgRNAs were cloned into the LentiCRISPR v2 vector using the BsmBI enzyme site. shRNA hairpins were cloned into pLKO.1 using AgeI and EcoRI enzyme sites. The human METTL1 catalytic domain, amino acids 160-163 (LFPD), was mutated to AFPA using PCR. Proteins were overexpressed by cloning them into the pLVX vector using EcoRI and BamHI sites. The cloned primers for sgRNA, shRNA and genes were listed in Supplementary Table 1.

## Senescence induction and SA-β-gal staining
After treating IMR90 cells with 10 μM etoposide (ETO) for 3 consecutive days, they were cultured for an additional 10 days to induce senescence. Replicative senescence: IMR90 cells were split at a ratio of 1:4. The cells reached senescence with growth arrest around passage 27. Senescence was confirmed through EdU incorporation and SA-βgal staining. Lentivirus-based inducible ER:KRasG12V was introduced into IMR90 cells to induce oncogene-induced senescence. Stable cells were selected by supplementing with puromycin for 3 days. Cells exhibited senescence 7 days after 4-Hydroxytamoxifen (4-OHT) treatment, leading to nuclear translocation of KRasG12V. Control cells were transduced with a vehicle.

SA-β-gal staining was conducted on in vitro cultured cells and tissue sections following the manufacturer's instructions. The mouse liver and kidney were fixed in 4% PFA in PBS overnight at 4 °C, followed by PBS washing, and then stored in 30% sucrose overnight at 4 °C. The tissues were embedded in optimal cutting temperature (OCT) compound, snap-frozen on dry ice, and stored at −80 °C. Frozen samples were sliced into 16 μm sections. After washing with PBS, sections were incubated with the SA-β-gal staining solution at 37 °C for 4 h (kidney tissue) or overnight (liver tissue). The quantification of SA-β-gal-positive cells was performed using ImageJ software.

## EdU incorporation assay
The EdU assay was performed following the manufacturer's instruction. IMR90 cells were briefly labeled with 1 mM EdU at 37 °C for 2 h and washed twice with PBS. The IMR90 cells were fixed with 4% paraformaldehyde (PFA) for 15 min at room temperature and then washed twice with 3% BSA in PBS. The cell was permeabilized in 0.3% Triton X-100 in PBS for 15 min, washed twice with 3% BSA-PBS, and then counterstained with 5 μg/ml DAPI for 10 min. The percentage of EdU-positive cells was calculated as the ratio of EdU-positive cell number to DAPI-positive cell number.

## Western blotting
The cells were harvested and lysed with RIPA lysis buffer containing EDTA-free Protease inhibitor cocktail (Roche Diagnostics, 5056489001), phosphatase inhibitor cocktail, and 1 mM PMSF for 30 min on ice. The lysis solution was sonicated for 20-30 cycles, protein concentration was assessed using a BCA kit, loading buffer with β-mercaptoethanol was mixed with the lysate, and the mixture was boiled for 5 min. Proteins were separated by 10%-12% SDS-PAGE gels and transferred to PVDF membranes based on their concentration.

After blocking PVDF membranes with 5% nonfat dry milk in TBST for 2 h at room temperature, they were incubated with the primary antibody overnight at 4 °C. After washing the membranes with TBST three times, they were incubated with an HRP-conjugated secondary antibody and detected using BeyoECL Star (Ultra hypersensitive ECL chemiluminescence kit) with the MINICHEMI™ Chemiluminescent Imaging System. The dilution of antibodies is followed by manufacturer guidance. Antibodies and reagents were listed in Supplementary Table 2 and Supplementary Data 6.

## RNA isolation, reverse transcription, and qRT-PCR
Total RNAs were extracted using Trizol reagent (Molecular Research Center, Cat# TR118) following the manufacturer's instructions. Reverse transcription was performed with Hiscript® III RT SuperMix for qPCR (+ gDNA wiper) with 1 μg total RNA in a 20 μl reaction system as per the manufacturer's instructions. 40 ng cDNA was used for gene expression analysis by qRT-PCR with 2x RealStar Green Fast Mixture (GenStar, Cat# A301-10) in the CFX Connect Real-time System (BIO-RAD). The primers for qRT-PCR assays are listed in Supplementary Table 3.

## Recombinant protein purification
Wild-type and D135S AlkB proteins were purified as previously described[41]. Widetype AlkB was amplified using E. coli genome DNA, and the AlkB-D135S mutant was generated by PCR. They were then cloned into the 6XHis-pET28A-Kan vector using BamHI/EcoRV. pET28a-AlkB and pET28a-AlkB-D135S were transformed into BL21(DE3) bacteria to produce recombinant proteins. Bacteria were inoculated and cultured in LB medium at 37 °C. When the OD reached 0.6–0.8, 1 mM IPTG and 5 μM FeSO₄ were added to the culture media to induce protein expression at 37 °C for 4 h. The bacteria were collected, lysed by sonication, and then centrifuged at 20,000 g at 4 °C for 90 min. The supernatant was collected for purifying recombinant proteins with Anti-His beads (QIAGEN) as per the manufacturer's instructions. The resulting protein was dissolved in a 10 mM Tris-HCl (pH 7.4) solution with 5% glycerol and stored at −80 °C for six months.

## mRNA purification and RNA decapping
Total RNA and mRNA contain a constitutive m7G cap. To quantify internal regulatory m7G, the cap was removed. Total RNA was isolated from cultured cells or tissues using Trizol. The mouse tissues were ground into powder using liquid nitrogen and then lysed with Trizol. The concentration was measured using Nanodrop. mRNA was purified from total RNA using the Dynabeads® mRNA DIRECT™ Kit protocol (Invitrogen) following the manufacturer's instructions. The m7G cap of mRNA was removed using RppH, as reported previously[41]. The reaction was prepared by combining a maximum of 5 μg total RNA in nuclease-free water with 5 μl of 10X Thermo Reaction Buffer, 2.5 μl of SUPERase-In RNase Inhibitor (Thermo Fisher Scientific) and 4 μl of RppH (NEB) in a final volume of 50 μl. The reaction was incubated at 37 °C for 2 h. Decapped RNA was extracted using RNA Clean & Concentrator (Zymo Research).

## tRNA purification
10 μg of total RNA was loaded with an equal volume of 2xTBE-urea loading buffer onto a 15% TBE-urea gel (Invitrogen) for tRNA separation. ssRNA bladder was used as a marker. The tRNA extraction involved cutting fragments ranging from 60 nt to 100 nt using the snap-frozen extraction method. The gel was pushed through a syringe into 600 μl of RNA extraction buffer (300 mM NaOAc, pH 5.5, 1 mM EDTA, 0.1U/ml SuperRNAsin). The mixture was incubated at 65 °C for 10 min with shaking, then snap-frozen in a gel slurry on dry ice or in liquid nitrogen. Thaw the gel slurry at 65 °C for 5 min with rotation. Elute RNA overnight or for 4–6 h at room temperature with gentle mixing on a rotating wheel, then filter through a Costar Spin-X

centrifuge tube. The RNA was precipitated with ethanol. The RNA pellet was dissolved in 20 µl of RNase-free water and stored at −80 °C for m7G detection or other purposes.

## Enzymatic RNA digestion for HPLC-MS

According to previously reported methods[87,88,102], RNA digestion was carried out. 200 ng total RNA or tRNA was denatured at 95 °C for 5 min using a PCR machine. The denatured RNA was then added to a digestion solution containing 0.5 µl nuclease S1, 2 µl 10x nuclease S1 buffer, and 0.5 µl was incubated at 37 °C for 2 h. Subsequently, 9 µl shrimp alkaline phosphatase (rSAP, NEB) and 4 ul of 10X buffer (50 mM tris-HCL, 10 mM MgCl2) were added and incubated at 37 °C for 2 h. The digested RNA solutions were filtered using 3 kDa MW cutoff ultra-filtration tubes (Pall, OD003C35). RNA concentration was measured using Nanodrop. The samples were diluted to 1 ng/µl using LC-MS grade water.

The digested total RNA, tRNA, and mRNA underwent HPLC-MS/MS analysis to detect m7G and G using an Agilent 1290 Infinity II system coupled with a 6495B triple quadrupole mass spectrometer (Agilent Technologies, Palo Alto, CA). A Waters C18 column (100 mm × 2.1 mm I.D., 1.8 µm particle size, Waters) was employed for HPLC separation of mononucleotides. The gradient elution profile was as follows: 0–1.0 min, 2.0% B; 1.0–5.0 min, 2–20.0% B; 5.0–6.0 min, 20-2% B. Solvent A contained 0.1% formic acid, while solvent B was 100% methanol. The mass spectrometer was operated in the positive ion mode. A multiple reaction monitoring (MRM) mode was adopted with the following transitions: 284-152 (G), 298.1-166.1 (m7G). The injection volume for each sample was 5 µl. The amounts of m7G and rG were calibrated with standard curves. MS was operated using a standard method.

## Northern blot

Northern blot was performed as previous report[103]. After mixing 5ug total RNA samples with 2X TBE loading buffer and incubating at 95°C for 5 min, the samples were loaded into a 10% TBE-UREA gel for RNA separation. After electrophoresis, the RNAs were transferred to a Hybond N+ membrane, treated with UV, and blotted with digoxigenin-labeled probes targeting tRNAs or U6 snRNA. The probe sequences are listed in Supplementary Table 4.

## Puromycin incorporation assay

The puromycin intake assay was performed as described previously[104]. For the in vitro assays, cells were treated with puromycin (1 µM) for 30 min. Cellular proteins were extracted using RIPA, followed by a western blot with anti-puromycin antibody (MABE343, Millipore). Actin was chosen as the control.

## Analysis for proteostasis by FlucDM-GFP reporter

Evaluation of proteostasis was performed by detecting aggregated FlucDM-GFP, as previously described[82]. In brief, lentiviral vector encoding FlucDM-GFP was transduced into IMR90 cells along with specified additional vectors in the experiments. Disrupted proteostasis is defined by the presence of more than three aggregated GFP inclusions per cell. The proteostasis index was evaluated by calculating the ratio of cells with aggregated GFP inclusions to the total number of GFP-positive cells.

## Chromatin immunoprecipitation (ChIP)-PCR and dual luciferase reporter assay

ChIP-PCR is performed to analyze SP1 protein binding to METTL1 promoter according to published protocol[105]. The primers used in ChIP-PCR are listed in Supplementary Table 5. The human METTL1 promoter region, spanning from the transcription start site (TSS) to −1000 base pairs, was cloned and inserted into the multiple cloning site (MCS) of the pGL3 basic vector using the Mlu I /Xho I (NEB)

enzyme, the primer was listed in Supplementary Table 1. Additionally, two SP1 binding sites within the promoter, "CGGGCGGACT" and "GGGGCGGGAA" were mutated to "ATTAATATTC" and "TTAACTA-TAG", respectively. To perform the dual luciferase reporter assay, either pGL3 basic or pGL3-p, along with the SP1 expression plasmid or an empty vector, and the Renilla luciferase plasmid, were co-transfected into 293 T cells using the PEI reagent. After a 24-h transfection period, luciferase activity was measured using a commercial reagent (Dual Luciferase Reporter Gene Assay Kit,Beyotime). Promoter activity was assessed by measuring luciferase activity and normalizing it with TK renilla activity.

## tRNA m7G TRAC-Seq

tRNA m7G TRAC-Seq was performed according to published protocol[30]. Small RNAs (<200 nt) were isolated using the mirVana miRNA Isolation Kit (Thermo Fisher Scientific). Small RNA was deacylated by removing 3' conjugated amino acids in a deacylation buffer (20 mM Tris-HCl, pH=9.0) at 37 °C for 45 min with a final concentration of 0.2 µg/µl. The deacylated RNA was precipitated using ethanol and glycoblue. The RNA was then incubated with recombinant wild-type and AlkB-D135S proteins to remove dominant methylations on RNAs, aiding reverse transcription. Briefly, 5 µg RNA was incubated with a 2× molar ratio of AlkB and a 4× molar ratio of AlkB (D135S) at 25 °C for 1 h in a reaction buffer containing 300 mM KCl, 2 mM MgCl2, 50 µM (NH4)2Fe(SO4)2·6H2O, 300 µM 2-ketoglutarate (2-KG), 2 mM L-ascorbic acid, 50 µg/ml BSA, and 50 mM pH 5.0 MES buffer. The reaction was quenched by adding 5 mM EDTA. The RNA was purified using Trizol reagent and precipitated with ethanol and glycoblue. Subsequently, half of demethylated RNA was used for small RNA sequencing library preparation (VAHTS Small RNA Library Prep Kit for Illumina, Vazyme) to analyze full-length tRNA abundance. The remaining half of the AlkB-treated RNA (approximately 2.5 µg) was treated with 0.1 M NaBH$_4$ for 30 min on ice in the dark, with 5 mM free m7GTP as a methylation carrier. The RNAs were precipitated by adding 0.1 volume of 3 M sodium acetate (pH 5.2) and 2.5 volumes of cold ethanol with glycoblue at −20 °C for 2 h. After precipitation, the NaBH$_4$-treated RNA was treated with an aniline-acetate solution (H2O: glacial acetic acid: aniline, 7:3:1) at room temperature in the dark for 2 h to induce site-specific cleavage[42,43]. The RNA was purified using the Oligo Clean & Concentrator kit (Zymo Research, D4060) and dissolved in 15 µl DEPC H2O for cDNA library construction using the VAHTS Small RNA Library Prep Kit for Illumina (Vazyme). Subsequently, dual-end sequencing was performed on the Illumina Nova 6000 platform.

## Ribosome nascent-chain complex-bound mRNA purification

Ribosome nascent-chain complex-bound mRNA purification was performed as described[56]. Three 15cm-dish cells were pre-treated with 100 µg/ml cycloheximide and incubated with 1 ml cell lysis buffer on ice for 30 min. Cell lysates were centrifuged at 16,200 g for 10 min at 4 °C. 10% of the extraction was used for input control. The remaining extraction was layered onto 35 mL sucrose buffer (30% sucrose in lysis buffer) and ultra-centrifuged at 174,900 g for 4 h at 4 °C using a SW32 rotor (Beckman Coulter, USA). The RNC pellets with polysome fractions were collected. Afterward, RNA samples were isolated from the input and RNC samples for qRT-PCR, with primer details provided in Supplementary Table 3.

## m7G-MeRIP-seq

m7G-MeRIP-seq is performed according to previous report with minor modifications[87]. Briefly, 4–6 µg human or mouse mRNA (two rounds of polyA+ purification) was fragmented into 50–100 nt using RNA Fragmentation Reagents (AM8740, Invitrongen) following the manufacturer's standard protocol. The fragmented mRNA was decapped with RppH or mRNA decapping kit (NEB) and then concentrated with RNA Clean & Concentrator (Zymo Research).

T4 Polynucleotide Kinase (EK0032, Thermo Fisher Scientific) was used to repair the end structures of these RNA fragments including 3'de-phosphorylation and 5'-phosphorylation, the repaired mRNA fragments 1 μg was kept as Input RNA, 2 μg repaired RNA were then incubated with 4 μL anti-m7G antibody (MBL) in 250 μL 1X IPP buffer (10 mM Tris-HCl, pH = 7.4; 150 mM NaCl; 0.1% NP-40) with freshly added 5% SUPERase-In RNase inhibitor (Thermo Fisher Scientific) at 4 °C for 2–4 h. Then 40 μL Dynabeads Protein G resins (Thermo Fisher Scientific) were washed twice with 1X IPP buffer, resuspended in 20 μL IPP buffer and added into the antibody-RNA mixture for another 2 h at 4 °C. The resins were then washed with 1X IPP buffer at 4 °C for four times. RNA was finally eluted with Proteinase K (recombinant, PCR grade, EO0491, Thermo Fisher Scientific) digestion. 45 μL 1X Proteinase K digestion buffer (2X recipe: 2% SDS, 12.5 mM EDTA, 100 mM Tris-Cl (pH = 7.4), 150 mM NaCl) with 5 mL Proteinase K was used to resuspend the resins, and the solution was incubated at 55 °C for 30 min. The second round of digestion was prepared in the same way and incubated at 55 °C for another 15 min. Both flow-throughs were combined, and m7G-containing RNA was recovered with RNA Clean & Concentrator (Zymo Research). Decapped and fragmented polyA+ RNA (as "input") and immunoprecipitated RNA (as "IP") were subjected to small RNA library construction (VAHTS Small RNA Library Prep Kit for Illumina, Vazyme). All libraries were sequenced on Illumina Nova 6000 platform with dual-end 80 bp read length.

## TRAC-Seq data and tRNA-Seq analysis

Raw TRAC-Seq reads were processed using Trim Galore v-0.6.7 (-length 25 -q 20) (https://github.com/FelixKrueger/TrimGalore) to remove adapter sequences and low-quality reads. Subsequently, the reads were aligned to the human mature tRNA sequences (hg38) downloaded from the GtRNAdb database[106], using Bowtie v-1.3.1 (-a -m 50 --best --strata) software, allowing for a maximum of three mismatches (-v 3)[107]. The read depth of each site on tRNAs was then calculated using bedtools (v-2.29.1)[108]. In addition, the m7G cleavage score can be determined to quantify the abundance of m7G modification[30,109]. The positions with a cleavage score >3 and a cleavage ratio >0.1 were considered as candidate sites for m7G. The cleavage score of the sites was calculated as in a previous report[30]. The motif surrounding identified m7G sites was determined through an unbiased motif search using the MEME v-5.4.0 software to verify the authenticity of the candidate tRNA m7G sites[110].

The human tRNA sequences (hg38) were downloaded from the GtRNAdb database[106]. Raw sequencing data must be preprocessed by removing sequencing adapters before analysis using the trimadapters.py program in tRAX[111]. The clean sequencing data were first mapped to the tRNA sequences using Bowtie2 (v-2.4.5, -k 100 --very-sensitive --ignore-quals --np 5)[112]. The read count for each tRNA was calculated using the "processsamples.py" program in tRAX. The raw counts were normalized using DESeq2(v-1.38.3)[113]. Read 1 sequencing data was used to perform the analysis; all tRNAs were covered by single-end data alone.

## Ribosome profiling (Ribo-Seq)

Ribosome profiling was conducted following a previously reported protocol[87]. Cycloheximide (CHX) was added to four 15-cm plates of IMR90 or MEF cells at 100 μg/ml and incubated at 37 °C for 2 min. The cells were harvested using a cell lifter with 5 ml of ice-cold PBS containing 100 μg/ml CHX, washed with a cold PBS-CHX solution, and the cell pellet was collected by centrifugation at 500 g for 5 min. 1 ml lysis buffer was used to suspend the cells, containing 10 mM Tris (pH 7.4), 150 mM KCl, 5 mM MgCl2, 100 μg/ml CHX, 0.5% Triton-X-100, and freshly added 1:100 protease inhibitor, 40U/ml SUPERasin. The solution was chilled on ice for 15 min with occasional pipetting and rotation. After centrifuging at 15,000 g for 15 min, the supernatant (~1 ml) was collected, and the absorbance was measured at 260 nm (150–200

A260 /ml). The total 'OD' of the lysate was calculated by multiplying the volume (in mL) by the A260 reading, typically requiring an OD of 10–20 for each experiment. The lysate was either processed immediately or flash-frozen in liquid nitrogen and stored at −80 °C for future use. Total RNA was extracted from 20% lysates. Around 200 μl of Portion I was extracted with 1.8 ml Trizol to obtain Input RNA. For the remaining 80% of lysates (Portion II) that were digested with ~26 U of RNase I (Ambion; EN0601) per OD for 1 h at room temperature (22 °C) with gentle agitation. To halt digestion, 10 μl of SUPERase•In™ RNase inhibitor was added to the reaction. Afterward, about 1 ml of digested ribosomes was loaded onto a 1 M sucrose cushion (prepared with lysis buffer without Triton-X-100) and centrifuged at 287000 g (Beckman, 345829) at 4 °C for 4 h, the pellet was resuspended in 30 μl of water, mixed with 1 ml of Trizol for RNA extraction, and finally dissolved in 10 μl of ddH2O. For size selection of ribosome protection fragments (RPF), 10 μl RNA was mixed with 10 μl 2 × TBE-urea loading buffer (Invitrogen) and run on a 10% TBE-urea gel. A 21-nt and a 42-nt ssRNA oligo served as size markers, and the gel band between 21 and 42 nt was isolated for mono-ribosome[114]. The gel was broken by passing it through a needle, and 600 μl extraction buffer (300 mM NaOAc, pH 5.5, 1 mM EDTA, 0.1 U/ ml SUPERase•In) was added. The gel slurry was heated at 65 °C for 10 min or longer with shaking, and then filtered through a 1 ml Qiagen filter or Costar Spin-X centrifuge tube filters. The RNA was purified using GlycoBlue and isopropanol. After repairing the RNA fragments from ribosome profiling (RPF) with T4 PNK, they were used to generate a library using the VATHS Small RNA Library Preparation Kit (Vazyme) and Ribo-off rRNA Depletion Kit (Human/Mouse/Rat) (Vazyme). All libraries were sequenced on Illumina Nova-Seq 6000 with dual-end 150 bp read length. The RNA was first purified using TRIzol, followed by RNA-Seq conducted by a commercial company.

## Data analysis for Ribo-Seq

Protein-coding gene sequences (hg38) and gene annotations (release-102) were obtained from the Ensembl Genomes database. The tRNA sequences were downloaded from the GtRNAdb database. The human rRNA sequences were obtained from the RPiso package (https://cosbi7.ee.ncku.edu.tw/RPiso/), while mtRNA, snoRNA, and snRNA sequences were retrieved from noncoding RNA annotations. The ribo-seq fastq file was processed to remove adaptors and low-quality reads using Trim Galore (-e 0.1 -q 30 --stringency 3). Reads between 25 and 34 nucleotides for Ribo-seq were kept for further analysis. When analyzing the data, single-end sequencing was sufficient for most ribosome-protected fragments (RPFs). The ribo-seq sequences were initially aligned to rRNAs, tRNA, snoRNA, and snRNA to eliminate contaminants using Bowtie (-L 20). Subsequently, the purified RPF sequences were aligned to the reference genome using STAR v-2.7.10a with specific parameters (--alignSJDBoverhangMin 1 --alignSJoverhangMin 51 --outFilterMismatchNmax 2 --alignEndsType EndToEnd --quantMode TranscriptomeSAM GeneCounts --outSAMattributes All)[115]. Ribo-seq quality assessment includes analyzing length distribution and trinucleotide periodicity (3-nt) using riboWaltz v-1.2.0[116]. The featureCounts program v-2.0.3 was utilized to count the RPFS uniquely mapped to CDS regions based on the genome mapping file (-t CDS -g gene_id)[117]. The codon occupancy of the A-site is determined by the frequency of in-frame P-sites within the coding sequence (26-33 nt) associated with each codon, calculated using riboWaltz software. The RNA-seq fastq file was performed to filter low-quality reads and adaptor sequences using Trimmomatic v-0.39[118]. RNA-seq reads were aligned using the default parameters of STAR for paired-end reads. The translation efficiency of genes was assessed by combining CDS RPF abundance and mRNA read abundance. The deltaTE method was then utilized to identify genes that were translationally regulated differently[119]. A false discovery rate (FDR) threshold value of 0.05 and a one-fold change were used as cutoffs to measure changes in the

 

translation efficiency of each gene. KEGG enrichment analysis of genes with reduced translation efficiency was performed using KOBAS27 (http://kobas.cbi.pku.edu.cn/kobas3/)[120].

### Data analysis for m7G-MeRIP

Adaptors and low-quality reads (length <20) were removed from raw sequencing reads using Trimmomatic software (v-0.39). Reads were aligned to the human genome (hg38) using the splice-aware STAR (v-2.7.10a) with parameters "--outFilterMultimapNmax 1". The high-confidence methylation sites were identified using exomePeak2 (v-1.10.0) with enrichment criteria of log2FC > 2, FDR < 0.05, and RPM > 1. ExomePeak2 was used for group differential peak analysis, comparing peak enrichment in shCtrl and shMETTL1 samples using parameters (log2FC > 1, FDR < 0.05).

### Reporting summary

Further information on research design is available in the Nature Portfolio Reporting Summary linked to this article.

## Data availability

The data supporting the findings of this study are available from the corresponding authors upon request. The raw sequencing data reported in this paper has been deposited in the Genome Sequence Archive in National Genomics Data Center, China National Center for Bioinformation/Beijing Institute of Genomics, Chinese Academy of Sciences that are publicly accessible at HRA006947 and HRA006075. The mass spectrometry proteomics data has been deposited to the ProteomeXchange Consortium via the iProX partner repository with the dataset identifier PXD052883. https://proteomecentral.proteomexchange.org/cgi/GetDataset?ID=PXD052883. Source data are provided with this paper.

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

## Acknowledgements

The authors thank Prof. Shuibing Lin at The First Affiliated Hospital of Sun Yat-sen University for providing Mettl1flox and Rosa26^LSL-Mettl1 (Mettl1 conditional knock-in) mice. We thank Prof. Demeng Chen at The First Affiliated Hospital of Sun Yat-sen University for his assistance with Northern blotting. This work was supported by grants from the National Natural Science Foundation of China (U20A2013), the National Key Research and Development Program of China (2018YFA0108100) and the Science and Technology Planning Project of Guangdong Province, China (2023B1212060050, 2023B1212120009). This work was also supported by the National Natural Science Foundation of China (T2222003, 32170849) and Frontier Research Program of Bioland Laboratory (2018GZR110105018).

## Author contributions

Yudong Fu performed the main experiments, collected, and interpreted data, and prepared the manuscript. Fan Jiang performed bioinformatic data analysis and interpretation, and prepared the manuscript. Yingyi Pan and Rui Xu performed *drosophila* experiments. Xiang Liang, Xiaofen Wu, Xiao Zhang, and Xingqing Li performed experiments or prepared reagents. Ruona Shi and Xiaofei Zhang helped to perform a proteomics assay.Xiao Zhang performed ChIP-PCR experiments.Dominique Ferrandon supervised *drosophila* experiments.Kaixuan Lin and Jing Liu helped to perform bioinformatics analysis. Jie Wang supervised the bioinformatic data analysis and interpretation, edited the manuscript; Tao Wang designed the study, supervised experiments, collected and interpreted the data, and wrote the manuscript. Duanqing Pei helped to improve the manuscript.

## Competing interests

The authors declare no competing interests.
