## [Peer Review File · Nature Communications]

Perturbation of METTL1-Mediated tRNA N7- Methylguanosine Modification Induces Senescence and AgeingREVIEWER COMMENTS

Reviewer #1 (Remarks to the Author):

The manuscript by Fu et al. reports a role for METTL1-mediated tRNA modification and protein translation control in suppressing cellular senescence of human fibroblasts and several aging phenotypes in mice. The authors found a decreased expression of an RNA methyltransferase, METTL1, in multiple models of cellular senescence and linked it to reduced levels of m7G modification of several tRNAs. METTL1 reduction is necessary and sufficient for cellular senescence. The authors suggest that METTL1 reduction leads to tRNA degradation, which in turn induces ribosome stalling and decreases protein translation. The authors then ramble between several mechanisms, including WNT signaling, ribotoxic stress response (RSR), and integrative stress response (ISR), to explain how METTL1 reduction modulates cellular senescence. Collectively, the authors propose a model in which METTL1 reduction causes ribosome stalling and decreases protein translation, which in turn contributes to cellular senescence and aging.

The limitation of the current study is that a reduction in protein translation does not seem to be universal in cellular senescence. Several previous studies showed that global protein translation is increased in senescent cells (Nat Cell Biol 2015, PMID: 26280535). Recent quantitative studies (Mol Cell 2022, PMID: 35987199; Front Cell Dev Biol 2022, PMID: 36133920) support this notion with a slightly different angle: Global protein synthesis increases but does not scale with cell size, promoting cellular senescence. These observations are consistent with the fact that the mTOR signaling, a positive regulator of protein synthesis, is constitutively active in different types of senescent cells (Cell Metab 2016, PMID: 27304503, J Cell Biol 2017, PMID: 28566325; J Cell Biol 2021, PMID: 33635313, Nat Metab 2023, PMID: 36864206). This discrepancy has limited enthusiasm for the manuscript at this time. Moreover, the current manuscript does not attempt to address how METTL1 (and eEF1A) reduction occurs during cellular senescence and aging, which may strengthen the role of METTL1-mediated m7G modification in cellular senescence and aging. Other specific concerns are detailed below:

1. The authors showed that METTL1 mRNA expression is reduced in senescent cells. However, the reduction of Mettl1 mRNA during mouse aging is only observed in liver and kidney, in contrast to the reduction of p16 mRNA, which was observed in all tissues examined (Figure S1C), suggesting that senescent cells accumulate not only in liver and kidney, but also in other tissues. This is odd and the authors need to address this issue.
2. In Figure 2D, the authors provided KEGG analysis of transcriptome from METTL1 KO cells. This reviewer assumes that these terms are enriched in upregulated genes upon METTL1 KO. Where are the terms related to SASP (e.g., inflammation)? This seems to be the opposite of what the authors observed in Figure 6.
3. Mettl1 KO greatly accelerates aging in mice (Figure 3 and S3), which seems to affect multiple tissues. However, the authors only examine several senescence markers in liver and kidney. Since this is a whole-body KO, the authors need to examine at least a few more tissues for senescence markers.
4. Cisplatin and Adriamycin treatment (CPA) induced cellular senescence in mice (Figure S3I) but did not reduce Mettl1 proteins. The authors need to explain this discrepancy.
5. Figures 4E and 4F do not match very well, despite the fact that METTL1 KO induces senescence. This suggests that m7G modification of tRNAs during senescence may be independent of METTL1 activity.
6. The authors showed that protein translation efficiency for genes involved in several biological processes is reduced upon METTL1 KO, which includes the p53 signaling pathway. This seems odd

because METTL1 KO induces cellular senescence and thus promotes, not inhibits, protein translation of the p53 signaling pathway, a positive regulatory pathway of cellular senescence.

7. In Figures 5F and 5G, the authors showed that WNT2 mRNA does not change during cellular senescence or upon METTL1 KO but its translation efficiency is reduced. The authors need to examine its protein levels during cellular senescence or upon METTL1 KO.

8. The authors suggested that eEF1A expression partially rescued cellular senescence caused by METTL1 KO, by protecting rapid tRNA decay. The authors should directly test this hypothesis by assessing the levels of some tRNAs, as shown in Figure 4I.

Reviewer #2 (Remarks to the Author):

In their manuscript, Fu and colleagues explore the relationship between METTL1-mediated m7G RNA modification and cell senescence. Building on the finding that the levels of the RNA methyltransferase METTL1 and its cofactor WDR4 are strongly reduced upon induction of cellular senescence, they show a corresponding decrease in the amount of m7G in both tRNAs and mRNAs. The effects on well-established senescence markers are effectively recapitulated in a METTL1 KO cell line where the authors also dissect the influence of expression of catalytically inactive METTL1. Using a mouse model, they generate data indicating that METTL1 contributes to maintaining longevity by preventing premature cell senescence. To uncover the mechanistic basis of the link between METTL1-mediated m7G methylation of tRNAs and senescence, the authors show reduced stability of hypomethylated tRNAs in the METTL1 KO cells and demonstrate that they can be recovered by overexpression of eEF1A. They also perform ribosome profiling and observe ribosome stalling at cognate codons. The inventory of transcripts differentially translated in cells lacking METTL1 include those encoding components of the Wnt signalling pathway, among them WNT2, a factor known to be a regulator of senescence. They further consolidate the impairment of Wnt signalling in the METTL1 KO cells and demonstrate that restoration of this pathway partially ameliorates senescence phenotypes.

While the findings of this study offer interesting insights into the functions of m7G modifications in tRNA in regulating cellular proliferation, a large proportion if not the majority of the sentences in the manuscript are grammatically incorrect and/or factually misleading, making the manuscript challenging to read. The inaccessibility of the text is further compounded by the fact that the motivations behind conducting particular experiments do not become apparent until conclusions are drawn.

If presented in a significantly more accessible form, this extensive study could provide an interesting contribution to the field. It addresses the impact of altered METTL1 expression on both the molecular and cellular levels using a broad portfolio of cutting-edge techniques. The data are generally convincing and the conclusions are largely well supported by the data. Beyond some experimental points listed below, the key issue is that the manuscript text requires significant improvement.

1. The whole text needs to be rewritten by a native English speaker or someone in excellent command of the language.
2. The motivations behind conducting particular experiments in several places do not become apparent until conclusions are drawn. For example, in line 206 treatment of mice with cisplatin and doxorubicin is described but only on line 219-220 does it become clear that the goal of this experiment was to determine whether METTL1 influences chemotherapy-induced senescence.
3. More consistency in the nomenclature used, especially in the figure labels, is also required. For example, METTL1 KO (Fig. 2D) and sgMETTL1 (Fig 2E) appear to refer to the same cell line. Likewise, in the complementation system, it seems that sometimes re-expression of METTL1 is labelled as "M1" and other times "METTL1", and the catalytically inactive mutant is termed "AFPA Mut", or "Mut" in

different panels. Many figure labels are also too small to read, e.g. scale bars on microscopy images and codons in graphs relating to the ribosome profiling data.

4. More attention also needs to be paid to explaining controls and marker proteins. For example, the "GFP" expressed in the complementation system is not explained and the use of EZH2, LMNB1, p16 and p21 as indicators of senescence is described long after western blots for these proteins are shown in the figures.

5. Upon expression of METTL1 or the catalytically inactive mutant in the METTL1 KO cell line, two bands are observed, neither of which appears to migrate at the same position as the endogenous protein detected in the control samples –the authors should clarify what exactly these bands correspond to.

6. Based on the western blots shown in Fig. 1B-E, it would be anticipated that WDR4 would also have been identified in their proteomic analysis of differentially expressed proteins in proliferation and senescence. Is this the case, and if yes, as a well-established RNA methyltransferase cofactor, it should be shown in Fig. 1A.

7. The original data of the HPLC-MS need to be shown in the supplementary data.

8. In Fig. 1F, the authors demonstrate that the proportion of m7G in mRNAs is substantially decreased, in fact more so than in their "tRNA" preparation. However, the impact of changes in mRNA m7G methylation on cellular senescence are not explored. The authors should analyse the available m7G maps and determine whether the modifications are enriched in transcripts encoding factors known to be linked to senescence. They should also compare these lists to the differentially translated mRNAs identified in their own work and, for any common targets, determine whether the extent of m7G methylation of these transcripts is altered in their senescence model systems.

Reviewer #3 (Remarks to the Author):

In this very interesting study, the authors explored how tRNA modification, N7-methylguanosine modification by METTL1/WDR4, regulates cell senescence and aging in cell and mouse models. The authors found decreased expression of METTL1 and WDR4 levels in senescent cells and in tissues (liver and kidney) from aged mice compared to young mice. A corresponding decrease in m7G levels measured by HPLC-mass spectrometry was found in the senescent cells and aged mouse tissue (Figure 1 and S1). Next the authors explored whether METTL1 plays any functional role in senescence by performing loss- and gain-of-function experiments in IMR90 human cells. Results showed that METTL1 deficiency promotes senescence whereas ectopic METTL1 (WT but not catalytic mutant) overexpression could suppress senescence in vitro (Figure 2 and S2). Next the authors examined the phenotypes of conditional METTL1 knockout (KO) and knock-in (KI) mice. Interestingly, they found METTL1 deletion in the adult mouse caused substantially increased cellular senescence and shortened lifespan with Mettl1 depleted mice developing striking degenerative and premature aging phenotypes. Conversely, they found that Mettl1 overexpression protects mice against chemotherapy-induced senescence (Figure 3 and S3).

To explore the molecular mechanisms underlying these interesting senescence phenotypes the authors examined changes in tRNA m7G modification and abundance using recently established sequencing methods and identified a set of thirteen m7G-modified tRNAs that are downregulated at both the m7G methylation and transcript level upon METTL1 KO mediated senescence (Figure 4 and S4). Ribo-Seq analysis revealed increased codon occupancy at codons that are decoded by m7G-modified tRNAs in METTL1-deficient cells, and an enrichment of genes in the WNT signaling pathway with decreased translation efficiency (Figure 5 and S5). Authors further report activation of stress response pathways downstream of altered translation and ribosome stalling in METTL1 deficient and senescent cells (Figure 6C and S6). Finally, it is shown that ectopic expression of eEF1A can (at least partially) some of the molecular effects of METTL1 deficiency (Figure 7 and S7).

Overall, this manuscript provides a large amount of high-quality data to support their original new findings and model whereby reduced levels of METTL1 during senescence lead to degradation of m7G-harboring tRNAs, causing ribosome stalling and perturbed mRNA translation. This affects key signaling pathways, including Wnt signaling, contributing to cellular senescence. Restoring certain proteins or activating Wnt signaling alleviates senescence effects, highlighting the role of tRNA modification in maintaining cellular homeostasis and preventing premature senescence.

The results are presented in a clear and logical way and the data support the authors conclusions. However, a few comments below should be addressed prior to publication in Nature Communications.

Specific comments:

1. The authors show quantification of m7G changes by HPLC-Mass Spectrometry which an appropriate quantitative method to measure nucleotide modifications. However, there appears to be some inconsistency between measurement in different figures. For example, Figure 1F, in control lane (proliferating IMR90 cells) the proportion of m7G:G in tRNA is $\sim 0.4\%$. However, in the control lane in Figure S2D (sgCtl IMR90 cells) this m7G:G ratio for tRNA is $\sim 1.6\%$. How can these percentages be so different between experiments in the same cell type? How were the tRNAs isolated for these measurements? This discrepancy raises some concerns about the accuracy of these measurements. Related to this, in other important figures (Figure S2F; Figure 3B) the authors use non-quantitative dot blots to measure relative m7G levels and these should be replaced with HPLC/MS measurements.
2. Figure 7. Authors show that eEF1A expression can suppress senescence, rescue the global translation phenotype, as well as the activation of downstream stress pathways in METTL1-deficient cells but they do not show that this also correlates with increased stability/expression of tRNAs. This data would help further support their model.
3. Figure 7E. It is not clear from the blot whether eEF1A expression rescues RPS20 Ubiquitination upon METTL1 knockdown.
4. The citations are a mess! So many of the references are replicated throughout the text (including #13/16; #36/38; #42/43; #39/44; #84/87; #31/85; etc..) and in the reference list. This needs to be fixed. Similar issues noted with the methods references.
5. Manuscript text could be improved with grammatical edits.

REVIEWER COMMENTS

Reviewer #1 (Remarks to the Author):

The manuscript by Fu et al. reports a role for METTL1-mediated tRNA modification and protein translation control in suppressing cellular senescence of human fibroblasts and several aging phenotypes in mice. The authors found a decreased expression of an RNA methyltransferase, METTL1, in multiple models of cellular senescence and linked it to reduced levels of m7G modification of several tRNAs. METTL1 reduction is necessary and sufficient for cellular senescence. The authors suggest that METTL1 reduction leads to tRNA degradation, which in turn induces ribosome stalling and decreases protein translation. The authors then ramble between several mechanisms, including WNT signaling, ribotoxic stress response (RSR), and integrative stress response (ISR), to explain how METTL1 reduction modulates cellular senescence. Collectively, the authors propose a model in which METTL1 reduction causes ribosome stalling and decreases protein translation, which in turn contributes to cellular senescence and aging.

The limitation of the current study is that a reduction in protein translation does not seem to be universal in cellular senescence. Several previous studies showed that global protein translation is increased in senescent cells (Nat Cell Biol 2015, PMID: 26280535). Recent quantitative studies (Mol Cell 2022, PMID: 35987199; Front Cell Dev Biol 2022, PMID: 36133920) support this notion with a slightly different angle: Global protein synthesis increases but does not scale with cell size, promoting cellular senescence.

These observations are consistent with the fact that the mTOR signaling, a positive regulator of protein synthesis, is constitutively active in different types of senescent cells (Cell Metab 2016, PMID: 27304503, J Cell Biol 2017, PMID: 28566325; J Cell Biol 2021, PMID: 33635313, Nat Metab 2023, PMID: 36864206). This discrepancy has limited enthusiasm for the manuscript at this time.

Moreover, the current manuscript does not attempt to address how METTL1 (and eEF1A) reduction occurs during cellular senescence and aging, which may strengthen

the role of METTL1-mediated m7G modification in cellular senescence and aging. Other specific concerns are detailed below:

Response: We appreciate the time and effort that the reviewers dedicated to providing valuable feedback and insightful comments on our manuscript. We have successfully implemented most of the suggestions provided by the reviewers.

In the study by Nat Cell Biol in 2015 (PMID: 26280535), an AHA incorporation assay was employed to quantify the total newly synthesized proteins in IMR90 ER:RAS cells after a 7-day treatment with OHT. The activity of SASP proteins was suppressed by the mTOR inhibitors rapamycin or torin1. The study concluded that inhibiting mTORC1 reduces the levels of SASP-related proteins in Ras-induced senescent cells. The investigation of protein translation and reduction in cellular senescence or aging was extensively carried out from the 1970s to the 1990s, as evidenced by studies cited in the following publications: PMID: 18346894; PMID: 36711035; PMID: 11718814. The data derived from a variety of organisms, including yeast, *C. elegans*, *Drosophila*, mice, rats, sheep, and humans, across a range of tissues including brain, lung, heart, thymus, muscle, liver, kidney, intestine, and pancreas, among others, provide evidence for the decline in protein translation rates with aging. This concept is consolidated in studies cited in PMID: 36711035, PMID: 6195209, PMID: 18199000, and PMID: 29545554. We have compiled the findings of these studies in Table 1, as outlined in the report (PMID: 36711035). Please refer to Table 1 for further details.

Table-1 Studies for Measurement of Protein Translation with Age (based on PMID: 36711035)

Species and Sex	Tissue	Translation Change Down	Age	Measure Method	Reference
C.elegans	Whole body	80%	Day2,5,8,12	35S-amino acid	PMID: 26865495
Drosophila	Whole body	30%	Day1-Day80	Radiolabeled mitochondria	PMID: 6425575
Drosophila	Head etc	Head 15%, Thorax 96%, Abdomen 33%	Day1/Day35	3H-amino acid	PMID: 119649
Drosophila	Whole Body	21%	Day1/Day55	3H-amino acid	PMID: 6415351
C57BL/6J ♀	Liver Kidney	Liver 45% Kidney 52%	Month3,6, 27	Radiolabeled mitochondria	PMID: 6425575
Mouse and Rat	Liver	Liver 36% (Mice), 43%(Rat)	various	Various	PMID: 1937398
C57BL/6J Mice ♀	Liver,brain Kidney,Muscles	Liver 65%, Brain 33% Kidney 70% Skeletal Muscle 85%	Month5/26	Postmitochondri preparations	PMID: 6350740
C57BL/6J Mice ♀	Liver,brain Kidney,Muscle	Peptide chain elongation decreased by 67% in brain, 80% in liver, 81% in kidney and 85% in sk muscle	Month 3-5/23-27	Cell-free preparations	PMID: 6738124
Fischer F344 rats	Brain	56%	Month6/32	Postmitochondria preparations	PMID: 7358282
Rat	Brain(forebrain)	20%	Month3/22.5	3H-L-lysine	PMID: 6778965
Mice	9 tissues	Liver 20%	Month3/18	Ribo-seq	PMID: 33264395
Yeast	cell	Global Reduction	Young/Senesc ent	Ribo-seq and Polysome profiling	PMID:30117416
Human, both sex	SK Muscle	Mitochondria 42%	24/54	13C-Leucine	PMID: 8986817

The studies (Molecular Cell 2022, PMID: 35987199; Front Cell Dev Biol 2022, PMID: 36133920; Cell 2019, PMID: 30739799; Science Advances, PMID: 34767451, and Cheng et al., 2021, BioRxiv) elucidated the impact of cell size on cellular proteasome composition. The proteasome is synthesized by the ribosome machinery within a cell. The impact of enlarged cell size (volume) on global mRNA translation and the functionality of ribosome machinery remains unclear.

mTORC1 activity remains consistently activated in senescent cells, as evidenced by studies published in reputable journals such as the Journal of Cell Biology in 2017 (PMID: 28566325), 2021 (PMID: 33635313), Nature Medicine in 2023 (PMID: 36864206), and Cell Metabolism in 2016 (PMID: 27304503). In the context of senescence characterized by constitutive mTORC1 activity, autophagy is suppressed. However, there is insufficient data available to determine whether there is an upregulation in global protein synthesis (J Cell Biol 2017, PMID: 28566325). The deletion of S6K1 resulted in an extended lifespan and enhanced resistance to age-related pathologies; however, the assessment of protein translation in these mice was not conducted (Science, 2009, PMID: 19797661). Indeed, there is a lack of evidence supporting the notion that administration of rapamycin in long term or the deletion of S6K1 in mice or cells leads to a reduction in protein translation (Nature Aging, 2023, PMID: 37142830; PMID: 23657975). Moreover, S6K1 KO mice have normal translational activity and respond normally to rapamycin, indicating that reduced translation plays a minimal role in the impact of chronic rapamycin administration on lifespan (PMID: 23839034, PMID: 27048303).

Senescent cells experience nucleolar stress and disruptions in rRNA processing and ribosome biogenesis due to the Rb pathway, as indicated by studies (PMID: 29941930 and PMID: 33077499). The studies mentioned above indicate that the overall protein translation status in senescent cells is influenced by both mTORC1 activity and the integrity of ribosome machinery.

Furthermore, numerous studies have clarified that mTORC1 selectively controls the translation of specific subtypes of mRNAs. In particular, the inhibition of mTORC1 in p53^{-/-} mouse embryonic fibroblasts predominantly affects mRNAs containing 5' terminal oligopyrimidine (TOP) motifs (Nature, PMID: 22552098; Nature Aging, PMID: 37142830).

Moreover, the inhibition of mTORC1 impedes the translation of numerous proteins; however, the translation efficacy of metabolic and stress-related genes is augmented in response to mTORC1 inhibition (Nature PMID: 23325216; Nature Aging PMID: 37142830).

Rapamycin, a compound that inhibits mTORC1, has been shown to reduce protein misfolding and aggregation according to studies cited in the following references: PMID: 15146184, PMID: 18199701, and PMID: 18636076. It was hypothesized that rapamycin might augment autophagy and decelerate mRNA translation. These mechanisms are believed to aid in the preservation of proteome homeostasis by facilitating the endogenous protein repair and degradation processes, thereby mitigating protein aggregation (PMID: 23325216).

In the revised manuscript, we try to clarify the potential impact of rapamycin on METTL1 function during senescence by treating METTL1 overexpressing cells with 10nM rapamycin. The data indicate no significant alteration in the senescence status, as determined by western blotting, in METTL1-overexpressing cells with or without rapamycin (Fig.S5H). Hence, our findings align with the established concept as reported in previous studies.

Our study focuses on the regulatory role of tRNA m⁷G methylation in maintaining protein translation homeostasis during senescence and aging. Disruption of mRNA translation via interference of METTL1 in proliferating cells has been shown to induce senescence (Fig. 2A-2E), whereas preserving tRNA m⁷G modification and mRNA translation may potentially delay cellular senescence (Fig.

2H-2M). Furthermore, in the revised manuscript, we demonstrate that the reduction in m7G modification hinders proteostasis. This is supported by the protein aggregation reporter FlucDM-GFP assay conducted in senescent cells and cells with depleted METTL1(Fig.7J-7K). Conversely, the replenishment of m7G levels could potentially alleviate protein aggregation that occurs during senescence (Fig.7J-7K).

In the revised manuscript, we address the mechanisms that underlie the decrease in METTL1 and eEF1A levels. The translation efficiency of eEF1A is observed to diminish in the absence of METTL1, as determined by RNC-QPCR analysis (Fig.7H). The data indicate a potential association between the downregulation of METTL1 and the decline in eEF1A levels.

To elucidate the regulatory mechanisms underlying the reduction of METTL1, an analysis of putative transcription factor binding sites in its promoters was performed using the PROMO procedure. The analysis revealed potential binding of the p53, c-Myc, E2F family proteins and SP1 to the METTL1 promoter. However, the expression level of METTL1 was found to be unaltered upon inhibition of these transcription factors using shRNA (Fig. S8A). The focus is on the sp1 protein, which interacts with the METTL1 promoter in cancer cells. The data presented in our study demonstrate the crucial role of the transcription factor SP1 in regulating the expression of METTL1, as depicted in Figure 8B. The chromatin immunoprecipitation polymerase chain reaction (ChIP-PCR) analysis demonstrated the binding of sp1 to the METTL1 promoter region at around -100bp. Moreover, our findings illustrate a reduction in the binding affinity of sp1 to the METTL1 promoter region as cells undergo senescence, as showed in Figure 8D. This observation is consistent with the concurrent reduction of METTL1 and SP1 levels in senescent cells, as illustrated in Figure 8A.

1. The authors showed that METTL1 mRNA expression is reduced in senescent cells. However, the reduction of Mettl1 mRNA during mouse aging is only observed in liver

and kidney, in contrast to the reduction of p16 mRNA, which was observed in all tissues examined (Figure S1C), suggesting that senescent cells accumulate not only in liver and kidney, but also in other tissues. This is odd and the authors need to address this issue.

Response: Thank the reviewer for their insightful and critical comments. Several additional tissues were included in the revised manuscript to evaluate the expression levels of METTL1 mRNA and protein (Fig.S1D). A notable reduction in METTL1 expression was observed in the liver, kidney, and lung tissues (Fig.1E, Fig.S1C, S1D), with no significant changes observed in other tissues like the small intestine and heart (Fig.S1C). Various organs are believed to be situated in distinct microenvironments, leading to different regulatory pathways for the aging process. Hence, it is plausible that the expression level of METTL1 may not be downregulated in all tissues. The function of METTL1 in these tissues may be affected by age-related factors, potentially through post-translational modifications or alternative signaling pathways.

2. In Figure 2D, the authors provided KEGG analysis of transcriptome from METTL1 KO cells. This reviewer assumes that these terms are enriched in upregulated genes upon METTL1 KO. Where are the terms related to SASP (e.g., inflammation)? This seems to be the opposite of what the authors observed in Figure 6.

Response: In the previous manuscript, not all enriched KEGG pathways were included in the main figures. In the revised manuscript, the SASP data is presented in Figure.S2D. Please review the figure.

3. Mettl1 KO greatly accelerates aging in mice (Figure 3 and S3), which seems to affect multiple tissues. However, the authors only examine several senescence markers in liver and kidney. Since this is a whole-body KO, the authors need to examine at least a few more tissues for senescence markers.

Response: In the revised manuscript, additional tissues such as lung, heart, and small intestine were collected for analyzing senescence markers. The data

indicated an expected increase in p21 and p16 levels in these tissues from knockout (KO) mice. Please refer to Fig.S3C-S3F for further information.

4. Cisplatin and Adriamycin treatment (CPA) induced cellular senescence in mice (Figure S3I) but did not reduce Mettl1 proteins. The authors need to explain this discrepancy.

Response: The administration of a low dose of Cisplatin and Adriamycin is expected to result in chronic damage to the liver and kidney tissues, leading to the appearance of senescent cells over an extended period. The samples in our previous manuscript were collected at an early time point, specifically 3 days after the first CPA treatment. In our revised manuscript, it was observed that the expression levels of METTL1 and WDR4 were downregulated after the third CPA treatment, as illustrated in Fig.S3J.

5. Figures 4E and 4F do not match very well, despite the fact that METTL1 KO induces senescence. This suggests that m7G modification of tRNAs during senescence may be independent of METTL1 activity.

Response: In our investigation, we have identified 20 tRNAs as substrates of METTL1. The reduction of METTL1 during senescence could potentially impact the levels of 20 transfer RNAs (tRNAs). Various mechanisms regulate tRNA levels, including transcriptional control by RNA polymerase III, DNA methylation, and histone methylation of tRNA promoters. For example, research has demonstrated that p53 and RB1 act to suppress tRNA transcription, whereas c-Myc promotes tRNA transcription (PMID: 12734418). Hence, it is anticipated that METTL1 deficiency plays a role in decreasing specific tRNA levels, while other pathways are involved in regulating other tRNA levels during senescence. So, we can observe that there is a lack of complete correspondence between Figure 4E and Figure 4F. We overlapped Fig. 4E and 4F to identify 13 tRNAs, defined as METTL1-regulated and senescence-associated tRNAs. Northern blotting confirmed that they are downregulated during senescence.

6. The authors showed that protein translation efficiency for genes involved in several biological processes is reduced upon METTL1 KO, which includes the p53 signaling pathway. This seems odd because METTL1 KO induces cellular senescence and thus promotes, not inhibits, protein translation of the p53 signaling pathway, a positive regulatory pathway of cellular senescence.

Response: Several proteins associated with the p53 signaling pathway were identified in the enriched list of translation efficiency (TE) analysis. This list includes CCND3, SIVA1, BAX, PIDD1, and BID, which were examined in the enriched p53 signaling pathway. The reduction in the expression of ccnd3 may induce senescence. SIVA1, BAX, BID, and PIDD1 are proteins known to play a role in the initiation of apoptosis. A reduction in the levels of these proteins may be linked to resistance to apoptosis in cellular senescence. Therefore, decreasing the translation of these proteins does not impact the regulatory role of the p53 signaling pathway in senescence.

7. In Figures 5F and 5G, the authors showed that WNT2 mRNA does not change during cellular senescence or upon METTL1 KO but its translation efficiency is reduced. The authors need to examine its protein levels during cellular senescence or upon METTL1 KO.

Response: The point is very helpful for the manuscript as it clarifies the regulatory function of METTL1 in Wnt2 translation. Following the reviewer's suggestion, we examined WNT2 protein level during cellular senescence and METTL1 KO, it was found a reduction in cellular WNT2 protein levels in both METTL1 KO IMR90 cells and replicative senescence IMR90 cells. For further details, please refer to Fig.5H.

8. The authors suggested that eEF1A expression partially rescued cellular senescence caused by METTL1 KO, by protecting rapid tRNA decay. The authors should directly test this hypothesis by assessing the levels of some tRNAs, as shown in Figure 4I.

Response 8: As per reviewer's suggestion, we performed Northern blotting to

analyze tRNA levels in eEF1A-expressing METTL1-KO cells. The study revealed that the overexpression of eEF1A had the capacity to inhibit the decrease of certain tRNA levels in the absence of METTL1. Please refer to Fig.7D.

Reviewer #2 (Remarks to the Author):

In their manuscript, Fu and colleagues explore the relationship between METTL1-mediated m7G RNA modification and cell senescence. Building on the finding that the levels of the RNA methyltransferase METTL1 and its cofactor WDR4 are strongly reduced upon induction of cellular senescence, they show a corresponding decrease in the amount of m7G in both tRNAs and mRNAs. The effects on well-established senescence markers are effectively recapitulated in a METTL1 KO cell line where the authors also dissect the influence of expression of catalytically inactive METTL1. Using a mouse model, they generate data indicating that METTL1 contributes to maintaining longevity by preventing premature cell senescence. To uncover the mechanistic basis of the link between METTL1-mediated m7G methylation of tRNAs and senescence, the authors show reduced stability of hypomethylated tRNAs in the METTL1 KO cells and demonstrate that they can be recovered by overexpression of eEF1A. They also perform ribosome profiling and observe ribosome stalling at cognate codons. The inventory of transcripts differentially translated in cells lacking METTL1 include those encoding components of the Wnt signalling pathway, among them WNT2, a factor known to be a regulator of senescence. They further consolidate the impairment of Wnt signalling in the METTL1 KO cells and demonstrate that restoration of this pathway partially ameliorates senescence phenotypes.

While the findings of this study offer interesting insights into the functions of m7G modifications in tRNA in regulating cellular proliferation, a large proportion if not the majority of the sentences in the manuscript are grammatically incorrect and/or factually misleading, making the manuscript challenging to read. The inaccessibility of the text is further compounded by the fact that the motivations behind conducting particular experiments do not become apparent until conclusions are drawn.

If presented in a significantly more accessible form, this extensive study could provide an interesting contribution to the field. It addresses the impact of altered METTL1 expression on both the molecular and cellular levels using a broad portfolio of cutting-edge techniques. The data are generally convincing and the conclusions are largely well supported by the data. Beyond some experimental points listed below, the key issue is that the manuscript text requires significant improvement.

On behalf of all the contributing authors, I would like to express our sincere appreciations of your constructive comments concerning our manuscript. These comments are all valuable and helpful for improving our article.

We have successfully implemented most of the suggestions provided by the reviewers. Following the recommendations, we have reviewed and edited the text throughout the entire manuscript. We hope that our answer meets the reviewers' requirements.

1. The whole text needs to be rewritten by a native English speaker or someone in excellent command of the language.

Response: We found the comments extremely helpful and have thoroughly reviewed and edited the text throughout the entire manuscript.

2. The motivations behind conducting particular experiments in several places do not become apparent until conclusions are drawn. For example, in line 206 treatment of mice with cisplatin and doxorubicin is described but only on line 219-220 does it become clear that the goal of this experiment was to determine whether METTL1 influences chemotherapy-induced senescence.

Response: Thank the reviewer for the suggestion. Following the suggestion, we described experiments design more clearly in the revised manuscript. The purpose of CPA treatment is to induce liver and kidney senescence mediated by DNA damage. Then we use the model to investigate whether METTL1 promote tissue recovery from senescence.

3. More consistency in the nomenclature used, especially in the figure labels, is also required. For example, METTL1 KO (Fig. 2D) and sgMETTL1 (Fig 2E) appear to refer to the same cell line. Likewise, in the complementation system, it seems that sometimes re-expression of METTL1 is labelled as “M1” and other times “METTL1”, and the catalytically inactive mutant is termed “AFPA Mut”, or “Mut” in different panels. Many figure labels are also too small to read, e.g. scale bars on microscopy images and codons in graphs relating to the ribosome profiling data.

Response: Thank the reviewer for the suggestions. Following the suggestion, we corrected the nomenclature by changing M1 to METTL1, Mut to AFPA mut and METTL1 KO to sgMETTL1. Scale bars and codons were enlarged in size. Please refer to the figures.

4. More attention also needs to be paid to explaining controls and marker proteins. For example, the “GFP” expressed in the complementation system is not explained and the use of EZH2, LMNB1, p16 and p21 as indicators of senescence is described long after western blots for these proteins are shown in the figures.

Response: Thank the reviewer for the suggestions. We corrected the sentence, please refer to 134-135, and 152-153.

5. Upon expression of METTL1 or the catalytically inactive mutant in the METTL1 KO cell line, two bands are observed, neither of which appears to migrate at the same position as the endogenous protein detected in the control samples –the authors should clarify what exactly these bands correspond to.

Response: Thank the reviewer for the suggestion. The two bands observed in the Western blotting represent the METTL1 protein. A FLAG tag was attached to the C-terminus of the METTL1 protein. We also detected two bands using a FLAG antibody in the Western blot experiments (Fig.S2F, right panel). It is possible that METTL1 may be deficient in a small portion at the N-terminus as a result of protease cleavage, leading to a modification in its migration pattern. To elucidate the matter, a Western blot analysis was conducted using METTL1 protein fused

with a HA tag at the N-terminus. The data indicate the presence of a single band (Fig. S2F, left), corresponding to the complete METTL1 protein. Our data indicates the presence of two forms of METTL1 within cells.

6. Based on the western blots shown in Fig. 1B-E, it would be anticipated that WDR4 would also have been identified in their proteomic analysis of differentially expressed proteins in proliferation and senescence. Is this the case, and if yes, as a well-established RNA methyltransferase cofactor, it should be shown in Fig. 1A.

Response: Thank the reviewer for the suggestion. I agree with the reviewer's perspective that WDR4 should be included in the proteome assay. Regrettably, our proteome assay was unsuccessful in obtaining reliable peptides originating from the WDR4 protein, primarily due to detection limitations. The formation of a heterodimer between METTL1 and WDR4 is essential for catalyzing m7G methylation. In preparation for proteome analysis, the samples were treated with a denaturing buffer, which resulted in the separation of the heterodimer into its individual free forms.

It is believed that a cell contains a cumulative copy number of 10^9 – 10^{11} protein molecules per cell (PMID: 27629639), which poses a challenge in detecting all of them in a single experiment. However, we detected the m7G catalytic enzyme METTL1, along with other RNA methylation enzymes and "readers" such as WTAP, YTHDC1, and TRMT1, in the proteome analysis. WDR4 reduction can be consistently detected during senescence and aging in the following experiments.

7. The original data of the HPLC-MS need to be shown in the supplementary data.

Response: Thank the reviewer for the suggestion. We added HPLC-MS data including standard curve, original values in supplemental figures, please refer to supplementary table 8 and source data for Fig.1F-G, Fig.S2E, Fig.S2G and Fig.3B.

8. In Fig. 1F, the authors demonstrate that the proportion of m7G in mRNAs is substantially decreased, in fact more so than in their “tRNA” preparation. However, the impact of changes in mRNA m7G methylation on cellular senescence are not explored. The authors should analyse the available m7G maps and determine whether the modifications are enriched in transcripts encoding factors known to be linked to senescence. They should also compare these lists to the differentially translated mRNAs identified in their own work and, for any common targets, determine whether the extent of m7G methylation of these transcripts is altered in their senescence model systems.

Response: Thank the reviewer for the suggestion. Quantification of m7G levels using HPLC-MS revealed that the tRNA-m7G level is roughly 40 times greater than that of mRNA-m7G. Consequently, our primary focus is on tRNA to elucidate the mechanisms underlying METTL1-mediated senescence. The role of mRNA-m7G in regulating senescence remains unclear. Following the reviewer's recommendation, m7G MeRIP experiments were performed on METTL1 knockout cells. The analysis of m7G-MeRIP data revealed that mRNAs showing reduced internal m7G methylation were linked to diverse pathways, including ribosome biogenesis, WNT pathway and degenerative diseases. This discovery is consistent with the differentially translated mRNAs identified in Ribo-seq. For further information, please refer to Fig.5J. The involvement of mRNA-m7G in senescence mediated by METTL1 deficiency, could be plausible to some extent.

Reviewer #3 (Remarks to the Author):

In this very interesting study, the authors explored how tRNA modification, N7-methylguanosine modification by METTL1/WDR4, regulates cell senescence and aging in cell and mouse models. The authors found decreased expression of METTL1 and WDR4 levels in senescent cells and in tissues (liver and kidney) from aged mice compared to young mice. A corresponding decrease in m7G levels measured by HPLC-mass spectrometry was found in the senescent cells and aged mouse tissue (Figure 1 and S1). Next the authors explored whether METTL1 plays any functional role in senescence by performing loss- and gain-of-function experiments in IMR90 human

cells. Results showed that METTL1 deficiency promotes senescence whereas ectopic METTL1 (WT but not catalytic mutant) overexpression could suppress senescence in vitro (Figure 2 and S2). Next the authors examined the phenotypes of conditional METTL1 knockout (KO) and knock-in (KI) mice. Interestingly, they found METTL1 deletion in the adult mouse caused substantially increased cellular senescence and shortened lifespan with *Mettl1* depleted mice developing striking degenerative and premature aging phenotypes. Conversely, they found that *Mettl1* overexpression protects mice against chemotherapy-induced senescence (Figure 3 and S3).

To explore the molecular mechanisms underlying these interesting senescence phenotypes the authors examined changes in tRNA m7G modification and abundance using recently established sequencing methods and identified a set of thirteen m7G-modified tRNAs that are downregulated at both the m7G methylation and transcript level upon METTL1 KO mediated senescence (Figure 4 and S4). Ribo-Seq analysis revealed increased codon occupancy at codons that are decoded by m7G-modified tRNAs in METTL1-deficient cells, and an enrichment of genes in the WNT signaling pathway with decreased translation efficiency (Figure 5 and S5). Authors further report activation of stress response pathways downstream of altered translation and ribosome stalling in METTL1 deficient and senescent cells (Figure 6C and S6). Finally, it is shown that ectopic expression of eEF1A can (at least partially) some of the molecular effects of METTL1 deficiency (Figure 7 and S7).

Overall, this manuscript provides a large amount of high-quality data to support their original new findings and model whereby reduced levels of METTL1 during senescence lead to degradation of m7G-harboring tRNAs, causing ribosome stalling and perturbed mRNA translation. This affects key signaling pathways, including Wnt signaling, contributing to cellular senescence. Restoring certain proteins or activating Wnt signaling alleviates senescence effects, highlighting the role of tRNA modification in maintaining cellular homeostasis and preventing premature senescence.

The results are presented in a clear and logical way and the data support the authors conclusions. However, a few comments below should be addressed prior to publication in Nature Communications.

Thank you for your positive comments and valuable suggestions to improve the quality of our manuscript. We appreciate the time and effort that the reviewer dedicated to providing valuable feedback and insightful comments on our manuscript.

Specific comments:

1. The authors show quantification of m7G changes by HPLC-Mass Spectrometry which an appropriate quantitative method to measure nucleotide modifications. However, there appears to be some inconsistency between measurement in different figures. For example, Figure 1F, in control lane (proliferating IMR90 cells) the proportion of m7G:G in tRNA is ~0.4%. However, in the control lane in Figure S2D (sgCtl IMR90 cells) this m7G:G ratio for tRNA is ~1.6%. How can these percentages be so different between experiments in the same cell type? How were the tRNAs isolated for these measurements? This discrepancy raises some concerns about the accuracy of these measurements. Related to this, in other important figures (Figure S2F; Figure 3B) the authors use non-quantitative dot blots to measure relative m7G levels and these should be replaced with HPLC/MS measurements.

Response: Thank you for the comment. Small RNAs (< 200nt) were purified using commercial kit from Invitrogen mirVana™ miRNA Isolation Kit or Macherey-Nagel™ NucleoSpin™ miRNA Kit .We replaced all dot blot result with HPLC-MS data, please refer to Fig.S2G, Fig.3B. The discrepancy in HPLC-MS data between measurements presented in different figures was resolved by ensuring that all measurements were conducted at the same facility.

2. Figure 7. Authors show that eEF1A expression can suppress senescence, rescue the

global translation phenotype, as well as the activation of downstream stress pathways in METTL1-deficient cells but they do not show that this also correlates with increased stability/expression of tRNAs. This data would help further support their model.

Response: Thank the reviewer for the point. The comment is beneficial for the successful completion of the study. Northern blot analysis was performed to assess the levels of m7G-tRNAs in METTL1-deficient cells with eEF1A overexpression. It is found that the m7G-tRNAs level partially recovered with eEF1A expression, as shown in Fig.7D.

3. Figure 7E. It is not clear from the blot whether eEF1A expression rescues RPS20 Ubiquitination upon METTL1 knockdown.

Response: Thank the reviewer for the suggestion. We conducted the experiment again. The revised findings indicate that the overexpression of eEF1A partially mitigated the ubiquitination of RPS20 following the knockout of METTL1, as illustrated in Fig.7F

4. The citations are a mess! So many of the references are replicated throughout the text (including #13/16; #36/38; #42/43; #39/44; #84/87; #31/85; etc..) and in the reference list. This needs to be fixed. Similar issues noted with the methods references.

Response: We are really sorry for our careless mistakes. Thank you for your reminder. We have addressed the problem by removing duplicated citations in the manuscript and methods section.

5. Manuscript text could be improved with grammatical edits.

Response: Thank the reviewer for the suggestion. The manuscript was enhanced by correcting grammatical errors and improving sentence structure following your suggestion.

REVIEWERS' COMMENTS

Reviewer #1 (Remarks to the Author):

In their revised manuscript, the authors have responded well to most of the major concerns raised by myself and other reviewers. I appreciate all the effort and dedication that the authors have put into their revision. However, I believe that the discrepancy between previous studies and the current study on changes in protein translation during cellular senescence is not satisfactorily resolved. The authors listed a number of papers suggesting that protein translation is reduced in various tissues in different species during aging (with the exception of one paper describing reduced protein translation during senescence in yeast). Yet, aging is not interchangeable with cellular senescence. In fact, senescent cells are a very small population even in most aged tissues, so it is hard to imagine that reduced protein translation during aging reflects reduced protein translation in senescent cells. Therefore, I suggest that the authors note several previous studies suggesting that global protein translation and the mTORC1 pathway are enhanced in senescent cells and discuss the potential discrepancy between them and the authors' current study on this matter. Senescent stimuli, cell type, or stage of senescence could be a factor for such a discrepancy.

Reviewer #2 (Remarks to the Author):

The reviewers have addressed most of the points raised by this reviewer. While the accessibility of the text has improved significantly and several aspects of the revision are satisfactory, especially the original points 3 and 7 remain to be appropriately addressed:

Original point 3: A more unified nomenclature helps the accessibility of the manuscript. The example mentioned in my original review (Fig. 2D and 2E) for different nomenclature used in different figures or panels for the same cell line has not been changed. As far as I understand both panels refer to the same cell line. Then this should indeed be checked throughout the manuscript and corrected.

Original point 7: The authors were asked to include the original data of the HPLC-MS measurements. They have now included the individual numbers of the data quantification, but not the MS profiles, which are important to see to evaluate data quality and the quantification. The actual MS profiles should be included as original data in the supplementary information.

After these amendments, the reviewer feels that the manuscript could be suitable for proceeding towards its publication.

Reviewer #3 (Remarks to the Author):

The authors have satisfactorily addressed all previous comments and this work is considered suitable for publication in Nature Communications.

REVIEWERS' COMMENTS

Reviewer #1 (Remarks to the Author):

In their revised manuscript, the authors have responded well to most of the major concerns raised by myself and other reviewers. I appreciate all the effort and dedication that the authors have put into their revision. However, I believe that the discrepancy between previous studies and the current study on changes in protein translation during cellular senescence is not satisfactorily resolved. The authors listed a number of papers suggesting that protein translation is reduced in various tissues in different species during aging (with the exception of one paper describing reduced protein translation during senescence in yeast). Yet, aging is not interchangeable with cellular senescence. In fact, senescent cells are a very small population even in most aged tissues, so it is hard to imagine that reduced protein translation during aging reflects reduced protein translation in senescent cells. Therefore, I suggest that the authors note several previous studies suggesting that global protein translation and the mTORC1 pathway are enhanced in senescent cells and discuss the potential discrepancy between them and the authors' current study on this matter. Senescent stimuli, cell type, or stage of senescence could be a factor for such a discrepancy.

Response: Thank you for your insightful comments and valuable feedback. Following your suggestions, we cited relevant reports on mTORC1 and discussed the inconsistency between protein translation levels and mTORC1 activity during senescence and aging. Please refer to manuscript lines 663-682. We appreciate the reviewer's consideration; the comment is helpful for maintaining the integrity of the manuscript.

Reviewer #2 (Remarks to the Author):

The reviewers have addressed most of the points raised by this reviewer. While the accessibility of the text has improved significantly and several aspects of the revision are satisfactory, especially the original points 3 and 7 remain to be appropriately addressed:

Original point 3: A more unified nomenclature helps the accessibility of the manuscript. The example mentioned in my original review (Fig. 2D and 2E) for different nomenclature used in different figures or panels for the same cell line has not been changed. As far as I understand both panels refer to the same cell line. Then this should indeed be checked throughout the manuscript and corrected.

Response: Thank you for your comments; they are helpful to us. We are sorry for any inconvenience caused by our inconsistent nomenclature. In the updated version of the manuscript, METTL1 mutant changed to AFPA-mut or AFPA-mutant. The sentence "All the senescence experiments were performed using IMR90 cells" was included in the Methods section.

Original point 7: The authors were asked to include the original data of the HPLC-MS measurements. They have now included the individual numbers of the data quantification, but not the MS profiles, which are important to see to evaluate data quality and the quantification. The actual MS profiles should be included as original data in the supplementary information.

Response: Thank you for your comments; they are helpful for us to improve the manuscript. We are sorry for our carelessness. We have included the product ion MS profile for G and m7G HPLC-MS data in Supplementary Data 7. Please find it.

After these amendments, the reviewer feels that the manuscript could be suitable for proceeding towards its publication.

Reviewer #3 (Remarks to the Author):

The authors have satisfactorily addressed all previous comments and this work is considered suitable for publication in Nature Communications.

Response: Thank you for your comment.

We would like to express our sincere gratitude to all reviewers for their effort and time in reviewing the manuscript. Your comments are very helpful to us.